# A fungal lytic polysaccharide monooxygenase is required for cell wall integrity, thermotolerance, and virulence of the fungal human pathogen *Cryptococcus neoformans*

**Corinna Probst[1,2☉], Magnus Hallas-Møller[3☉], Johan Ø. Ipsen[3], Jacob T. Brooks[4], Karsten Andersen[3], Mireille Haon[5], Jean-Guy Berrin[5], Helle J. Martens[3], Connie B. Nichols[1,2], Katja S. Johansen[3], J. Andrew Alspaugh[1,2]\***

1 Department of Medicine, Duke University School of Medicine, Durham, North Carolina, United States of America, 2 Department of Molecular Genetics and Microbiology, Duke University School of Medicine, Durham, North Carolina, United States of America, 3 Department of Geoscience and Natural Resource Management, University of Copenhagen, Frederiksberg, Denmark, 4 Department of Physics and Astronomy Department, University of North Carolina at Chapel Hill, Chapel Hill, North Carolina, United States of America, 5 INRAE, Aix-Marseille Univ., Biodiversité et Biotechnologie Fongiques (BBF), Marseille, France

☉ These authors contributed equally to this work.
* andrew.alspaugh@duke.edu

**Data Availability Statement:** All data is present in the Manuscript files.

## Abstract

Fungi often adapt to environmental stress by altering their size, shape, or rate of cell division. These morphological changes require reorganization of the cell wall, a structural feature external to the cell membrane composed of highly interconnected polysaccharides and glycoproteins. Lytic polysaccharide monooxygenases (LPMOs) are copper-dependent enzymes that are typically secreted into the extracellular space to catalyze initial oxidative steps in the degradation of complex biopolymers such as chitin and cellulose. However, their roles in modifying endogenous microbial carbohydrates are poorly characterized. The *CEL1* gene in the human fungal pathogen *Cryptococcus neoformans* (*Cn*) is predicted by sequence homology to encode an LPMO of the AA9 enzyme family. The *CEL1* gene is induced by host physiological pH and temperature, and it is primarily localized to the fungal cell wall. Targeted mutation of the *CEL1* gene revealed that it is required for the expression of stress response phenotypes, including thermotolerance, cell wall integrity, and efficient cell cycle progression. Accordingly, a *cel1Δ* deletion mutant was avirulent in two models of *C. neoformans* infection. Therefore, in contrast to LPMO activity in other microorganisms that primarily targets exogenous polysaccharides, these data suggest that *Cn*Cel1 promotes intrinsic fungal cell wall remodeling events required for efficient adaptation to the host environment.

## Author summary

Fungi need to adapt quickly to environmental stresses to thrive. The fungal cell wall, which supplies support and integrity to the cell, is an essential compartment to react and

**Funding:** Scanning electron microscopy, performed at the Chapel Hill Analytical and Nanofabrication Laboratory (CHANL), was supported by the National Science Foundation (award number ECCS-1542015). This work was also supported by funding from the National Institute of Health [NIGMS R01GM041840 (JAA)] and the Novo Nordisk Fond [NNF17SA0027704 (KSJ) and NNF21OC0070070 (KSJ and JAA)]. The funders had no role in study design, data collection and analysis, decision to publish, or preparation of the manuscript.

**Competing interests:** The authors have declared that no competing interests exist.

interact with the surrounding environment. Rapid changes within the carbohydrate composition and architecture occur in response to environmental stresses. Lytic polysaccharide monooxygenases (LPMOs) are mononuclear copper-enzymes, secreted by microbes to assist in the first steps of remodeling and degradation of complex and recalcitrant carbohydrates. In this study we explore the role of the putative AA9 family LPMO *Cn*Cel1 for growth and virulence of the fungal human pathogen *Cryptococcus neoformans*. The *CEL1* gene is highly up-regulated in presence of host stresses and a *cel1Δ* mutant strain is avirulent in a murine model of infection. Downstream analysis of virulence-associated phenotypes identified the *CEL1* gene to be required for thermotolerance as well as cell wall integrity, and efficient cell cycle progression in presence of host-mimicking stresses. Based upon those findings, we propose that *Cn*Cel1 likely promotes intrinsic fungal cell wall remodeling events essential for adaptation to the host environment.

## Introduction

Complex carbohydrates serve diverse roles in microbial structure, metabolism, and interaction with the external environment. A unique subset of these complex polysaccharides is present in the fungal cell wall, a cell surface structure composed of ordered layers of inter-linked carbohydrates such as chitin, chitosan, and glucans [1]. As an important part of stress adaptation, the fungal cell wall undergoes extensive modification in response to extracellular cues [2]. For fungal pathogens, these include host-derived signals such as elevated pH and oxidative stress, changes in osmolarity and temperature, and alterations in metal availability [2–4].

Within the infected host, the cell surface is actively remodeled to adapt to changes in the external environment, including promoting intermolecular cross-linking to strengthen the cell wall, as well as altering the polysaccharide content to embed more immunogenic cell wall epitopes deeply in the cell wall structure while expressing surface components that promote immune evasion [2,5–7]. Many enzymes involved in the biosynthesis of specific cell wall polysaccharides are well defined, including α- and β-glucan synthases, chitin synthases, and chitin deacetylases [1]. However, the networks of predicted transglycosylases, that promote further cell wall structural complexity, and carbohydrate degrading enzymes, that allow cell wall plasticity during stress, are less well characterized [8].

*Cryptococcus neoformans* (*Cn*) is a human fungal pathogen that predominantly causes infection among highly immunocompromised patients, including those with advanced HIV infection [9,10]. Resulting in a lethal form of fungal meningitis, this neuropathogen causes an estimated 170,000 deaths each year in vulnerable patients with AIDS [11]. Host-associated signals help to direct modifications of the *Cn* cell wall that promote the efficient binding of an additional cell surface feature, the polysaccharide capsule. Composed of branching polymers such as glucuronoxylomannan (GXM), the cryptococcal capsule is an inducible structure that is required for pathogenesis [6,12,13].

Lytic polysaccharide monooxygenases (LPMOs) are secreted enzymes that cleave complex polysaccharides such as cellulose, chitin, and starch [14,15]. LPMO genes are widely present in microbial genomes, likely reflecting the common need to degrade carbohydrates encountered in the environment that are recalcitrant to the action of polysaccharide hydrolases and other modifying enzymes [16]. LPMOs are divided into several families based on their protein sequences [17]. The *Cn* genome contains three predicted LPMO genes from different families: one AA9 (CNAG_00601/*CEL1*), one AA11 (CNAG_03405) and one AA14 (CNAG_07314). The predicted substrate specificities suggest that the AA9 and AA14 would cleave plant-based

polysaccharides (AA9: cellulose, AA14: substrate unknown), while the AA11 might cleave fungal or insect-based polysaccharides (i.e., chitin). A previously reported *C. neoformans* LPMO-like protein, *Cn* Bim1/Cbi1, has been subsequently shown not to possess LPMO activity [18]. In contrast, this protein, also known as X325 in other fungi and structurally similar to LPMOs, promotes copper homeostasis by binding and releasing copper from the cell wall in response to oxidative signaling and metal availability [4,19,20].

Interestingly, a transcriptomic study of *Cn* clinical strains demonstrated induction of the AA9 and AA14 LPMO genes in two cases of human cryptococcal meningitis. The induction of the AA14 gene was strain-specific during infection, although the "non-induced" strain had a high baseline expression of AA14 on rich medium *ex vivo* [21]. These results suggest that *Cn* LPMOs might function during pathogenesis in the human host, despite there being neither cellulose nor xylan in that environment. Recently, other investigators determined that a "chitin-active" LPMO promoted virulence in *Pseudomonas aeruginosa*, likely by its effects in modifying cross-linked peptidoglycans in the bacterial cell wall [22,23]. We therefore hypothesized that the *Cn* LPMOs might act on endogenous fungal substrates, such as the polysaccharides within the cryptococcal cell wall and capsule.

To test this hypothesis, we assessed the physiological consequences of mutations in the *CEL1* gene encoding a predicted fungal-specific LPMO of the AA9 family. These mutations included a complete gene deletion, as well as targeted disruption of the postulated Cu-binding histidine brace. Physiological and cell wall changes in the *cel1* mutants *in vitro* suggest that the *Cn*Cel1 protein is involved in endogenous cell wall polysaccharide modifications, rather than in the decomposition of substrates in the environment for uptake and further sugar metabolism. This function was dependent on an intact His-His brace. The *Cn*Cel1 protein is specifically induced in response to the neutral/slightly alkaline pH of the infected host, and this induction requires intact function of the alkaline-response Rim signaling pathway. Moreover, *Cn*Cel1 is required for full virulence in two models of *Cn* infection. These studies indicate a new function for fungal LPMOs to control resistance to cell wall, oxidative, and thermal stresses, and thereby to promote structural and metabolic adaptability during infection.

## Results

### CNAG_00601 encodes a polypeptide that shares features with LPMO mononuclear copper enzymes

Three predicted LPMO genes are present in the *C. neoformans* var. *grubii* (strain H99) genome. Phylogenetic comparison with other LPMO sequences in the carbohydrate-active enzyme (CAZy) database reveals that the CNAG_00601 gene product, *Cn*Cel1, shares most homology with AA9 LPMOs, CNAG_03405 most closely aligns with the AA11 family, and CNAG_07314 likely encodes an AA14 family protein (Fig 1A). Analysis of *C. neoformans* transcriptional activity during infection revealed induction of CNAG_00601 in all tested strains [21]. This putative AA9 LPMO-encoding gene was therefore chosen for further investigation.

Upon cleavage of its N-terminal signal peptide, the mature protein encoded by CNAG_00601 is predicted to include 375 amino acids (S1A Fig). The resulting N-terminal residue of *Cn*Cel1 is a conserved histidine residue in LPMOs likely involved in formation of the copper histidine brace [24]. Structure-based alignment with similar fungal LPMOs suggests that *Cn*Cel1 possesses the conserved in-space tyrosine residues involved in AA9 cellulose binding (Tyr24 and Tyr212 in *Ta*AA9A), however *Cn*Cel1 differs from other AA9s by having additional aromatic residues in the L2 and LC regions (Trp35, 234, 238 and Tyr27 in *Cn*Cel1), which could alter the substrate recognition from classical cellulose-active AA9 LPMOs, e.g., *Ta*AA9A (Figs S1B, 1B and 1C). Additionally, *Cn*Cel1 shows a long C-terminal extension

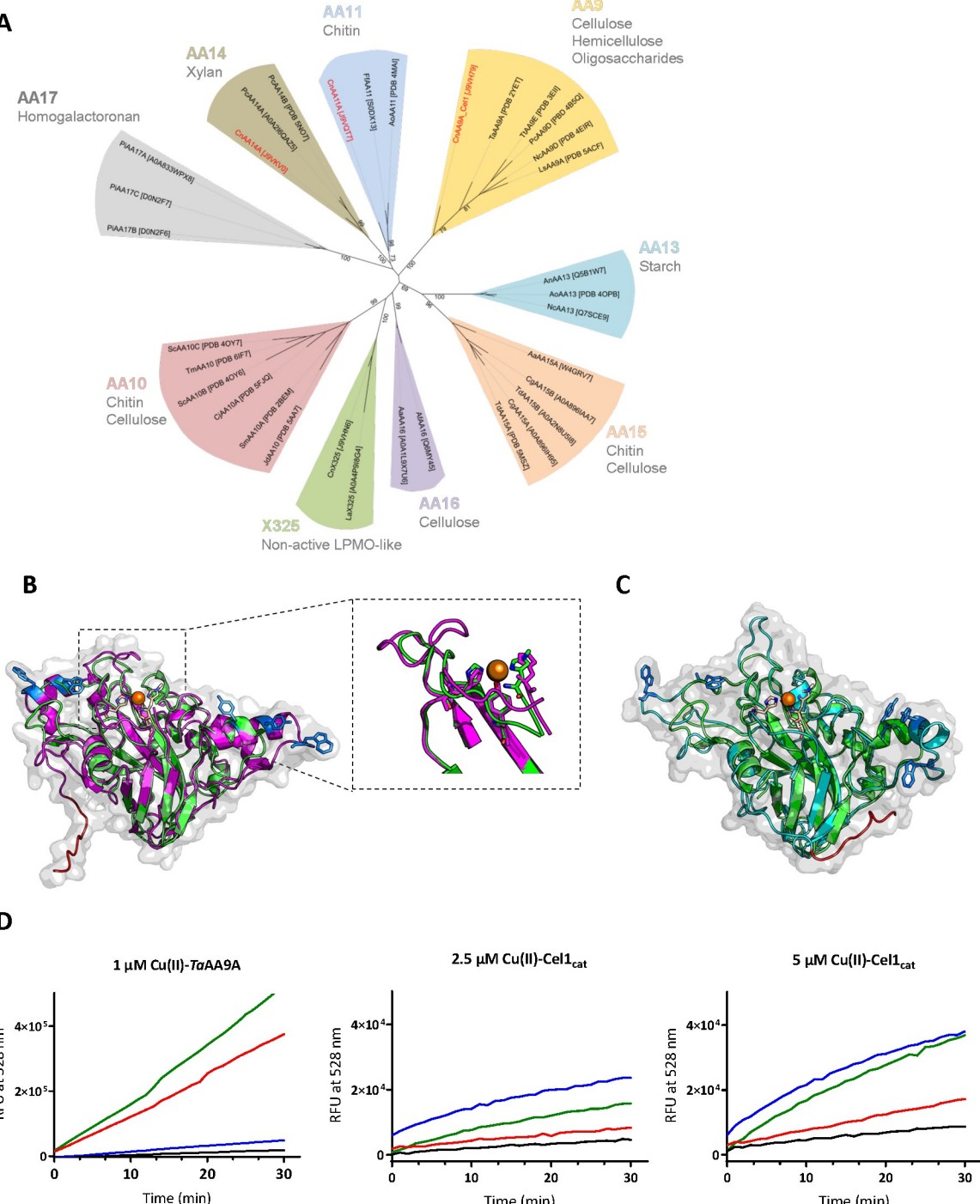

**Fig 1. *Cn*Cel1 shows homology with LPMOs of the AA9 family. (A)** Phylogenetic tree of lytic polysaccharide monooxygenase (LPMO) families. Sequences of characterized proteins only (Uniprot number or Protein Data Bank (PDB) code provided in the figure) were aligned using T-coffee Expresso [89]. Phylogeny was inferred using PhyML software [86] and the Whelan and Goldman (WAG) amino acid substitution model [87]. Branch support was calculated by 100 bootstrap repetitions (value displayed on tree). The tree was visualized with Interactive Tree Of Life (iTOL) software [88]. Substrate specificities of each LPMO family are shown in grey. **(B)** Model of *Cn*Cel1 (magenta) made using alphafold2 and downloaded from Uniprot [91]. The model is aligned to TaAA9A (PDB: 2YET) (green). The modelled surface of *Cn*Cel1 is shown in light grey. *Cn*Cel1 aromatic residues potentially involved in substrate binding are shown as stick representation (blue). The histidine brace and the apical tyrosine align. The *Cn*Cel1 C-terminal region is shown shortened and in red **(C)** Phyre2 model of *Cn*Cel1 (cyan). The *Cn*Cel1 surface is shown in gray. The model is aligned to TaAA9A (PDB: 2YET) (green). *Cn*Cel1 aromatic residues potentially involved in substrate binding are shown as stick representation (blue). The *Cn*Cel1

dCTR region is shown shortened and is highlighted in red. (**D**) Progress curves for oxidation of reduced fluorescein by Cel1$_{cat}$. As positive control the oxidation of reduced fluorescein by 1 μM copper loaded-TaAA9A is shown. Activity measured by relative fluorescent units (RFU) at 528 nm for 30 min. Reaction conditions are copper loaded enzyme concentrations as indicated, 75 mM phosphate citrate, pH 7.4, 25˚C (black line). 100 μM dehydroascorbic acid (red line), 100 μM H$_2$O$_2$ (blue), or both (green line) were added as co-substrates to investigate Cel1$_{cat}$ redox cycling propensity. Experiments done in triplicates, standard deviations shown, but not visible.

(S1 Fig). Similar elongated and intrinsically disordered C-terminal regions (dCTRs) are present in many LPMOs from divergent organisms, but their roles are currently unknown [25].

The predicted three-dimensional structure of the protein encoded by CNAG_00601 was modeled using both alphafold2 and phyre2 based upon the *Ta*AA9A structure from *Thermoascus aurantiacus* (PDB: 2YET). Both the overall folding of the *Cn*Cel1 protein and the predicted copper-active site were found to be in acceptable agreement with the *Ta*AA9A protein (Fig 1B and 1C). The sequence of *Cn*Cel1 has six cysteines, and both the structure-based alignments and the two folding models fail to pair two of the cysteines (S2A Fig). Therefore, these models can only serve as a best estimate. Only minor differences between the phye2 model and an alphafold2 model provided by the Uniprot database were observed (Fig 1B and 1C).

According to LPMO protein convention, the gene product of CNAG_00601 should be named *Cn*AA9. However, this gene was formerly noted to be upstream of the *CAP60* gene, and it was previously named *CnCEL1* based on homology to the coding region of enzymes that were at the time mis-classified as cellulose-active glycosyl hydrolases in other fungal species [26]. We maintained the established nomenclature, although the specific activity of *Cn*Cel1 is not yet known.

To further investigate whether *Cn*Cel1 is capable of LPMO activity *in vitro*, the *Cn*Cel1 catalytic domain (Cel1$_{cat}$) was recombinantly produced in *Pichia pastoris* (syn. *Komagataella phaffii*) without its unstructured C-terminus region (dCTR, S1A Fig). The enzyme was purified, and the amino acid sequence confirmed by mass spectrometry. The enzyme was deglycosylated by EndoH treatment and the concentration determined using quantitative amino acid analysis (S2B–S2D Fig).

The native substrate of *Cn*Cel1 is unknown, therefore to assess if the enzyme was functional and able to redox cycle between Cu$^{2+}$- and Cu$^{1+}$- states, a substrate free assay, based on a recently published protocol, was used to measure Cel1$_{cat}$ redox activity [27]. This assay is based upon the ability of LPMOs to oxidize reduced fluorescein. As shown in Fig 1D, Cel1$_{cat}$ oxidized reduced fluorescein in a time- and concentration-dependent manner, characteristics of an enzyme. In agreement with its predicted copper dependency, Cel1$_{cat}$ activity further increased when the Cel1$_{cat}$ enzyme was copper-loaded prior to assay. The copper loading ensured copper saturation of the active sites. Free copper was tested in assay conditions to control for background activity (Figs S2E and 1D). In contrast to the well characterized cellulose-active LPMO *Ta*AA9A, the increase in reaction rate of Cel1$_{cat}$ was substantially increased with the addition of hydrogen peroxide (Fig 1D).

## *CEL1* expression is induced under host-like conditions in a Rim pathway-dependent manner

To explore *Cn*Cel1 expression in response to specific host-associated conditions, we measured protein abundance *in vitro* after changes in pH, copper availability, and temperature. We created a *C. neoformans* strain replacing the wild-type *CEL1* locus with a *CEL1-4xFLAG* allele expressing a full length, C-terminally epitope-tagged Cel1 protein under control of its endogenous promoter. This strain was phenotypically similar to the wild-type strain, and not to the *cel1*Δ mutant strain, in all tested *in vitro* conditions, indicating that this fusion protein was

likely fully functional (S3A Fig). Cel1-4xFLAG expression was specifically induced at pH 8 compared to more acidic pH values (Fig 2A). Analysis of previously published RNA-Seq data-sets identified the *CEL1* transcript to be up-regulated in response to alkaline pH stress in a Rim101-dependent manner [28,29] (Fig 2B). The *Cn* Rim pathway is required for the cellular response to alkaline pH [30]. Consistent with this transcript data, we observed a time-dependent induction of the Cel1-4xFLAG protein expression after shift from synthetic complete media (SC) pH 5 to pH 8, with maximal expression by 8 hours and slight reduction by 24 hours (Fig 2C). However, no induction of Cel1-4xFLAG expression was observed in a *rim101Δ* mutant strain (Fig 2C).

Based on published transcript data [28,29], Rim101-dependent induction at alkaline pH is specific for *CEL1*, and not for the genes encoding the other two putative *Cn* LPMOs (CNAG_03405 and CNAG_07314, Fig 2B). Furthermore, no co-regulation was observed in these transcriptional datasets between *CEL1* and the capsule-regulating *CAP60* gene, which is located immediately downstream and transcribed in the same direction as the *CEL1* gene [26] (Fig 2B).

Although *Cn*Cel1 is predicted to be a copper-binding protein based on its sequence similarity to LPMOs, no changes in expression were observed after treatment either with excess copper (1mM CuSO4) or with copper sequestration by the extracellular copper (I) chelator bathocuproine disulfonic acid (BCS, 1mM) (Fig 2A). Additionally, no measurable Cel1-4xFLAG induction was detected by western blot during cold (4°C) or high-temperature (37°C) stress at pH 5 (Fig 2A). However, we observed temperature-dependent induction of Cel1-4xFLAG expression at alkaline pH. When *Cn* was incubated in SC buffered to pH 8.0, there was no detectable Cel1-4xFLAG protein during incubation at 4°C, but there was a temperature-dependent increase of protein expression from 22°C to 39°C (Fig 2D). Therefore, two conditions strongly associated with mammalian infection, slightly alkaline pH and elevated temperatures, resulted in an additive induction of *Cn*Cel1 protein levels. Similarly, we observed a strong induction of Cel1-4xFLAG protein during incubation in $CO_2$-Independent Medium (CIM) at 37°C to mimic host-like conditions *in vitro* (Fig 2E).

## *Cn*Cel1 is not secreted extracellularly but localizes to the *Cn* cell surface

LPMOs are typically secreted, extracellular enzymes [16]. The predicted full-length *Cn*Cel1 protein includes an N-terminal signal peptide, suggesting that *Cn*Cel1 is likewise a secreted, extracellular enzyme (S1A Fig). However, western blot analysis of the Cel1-4xFLAG protein after 24h induction in tissue culture medium only detected the tagged protein in the cell pellet and not secreted into the growth medium (Fig 2F). We therefore hypothesized that *Cn*Cel1 possibly remains associated with the *Cn* cell surface. To test this hypothesis, we incubated Cel1-4xFLAG expressing cells with the Zymolyase yeast lytic enzyme mix to digest the fungal cell wall and thus to remove all cell wall-associated proteins from the *Cn* cell surface (Fig 2G–2H). No changes of the intracellular histone 3 (H3) protein or the ER/Golgi resident, GFP tagged Sec63 protein (SEC63-GFP) was detected upon Zymolyase treatment [31]. In contrast, a dose dependent decrease of the Cel1-4xFLAG signal was observed when cells were treated with increasing concentration of the Zymolyase yeast lytic enzyme mix, indicating that Cel1-4xFLAG is likely associated with the *Cn* cell wall.

## *Cn*Cel1 is required for virulence in two models of *Cn* infection

To assess the requirement of *Cn*Cel1 for infection pathogenesis, we created a *cel1Δ* loss-of-function mutant and explored fungal-host interactions using two independent models of cryptococcal infection. We also complemented the *cel1Δ* mutant by reintroducing a WT copy of

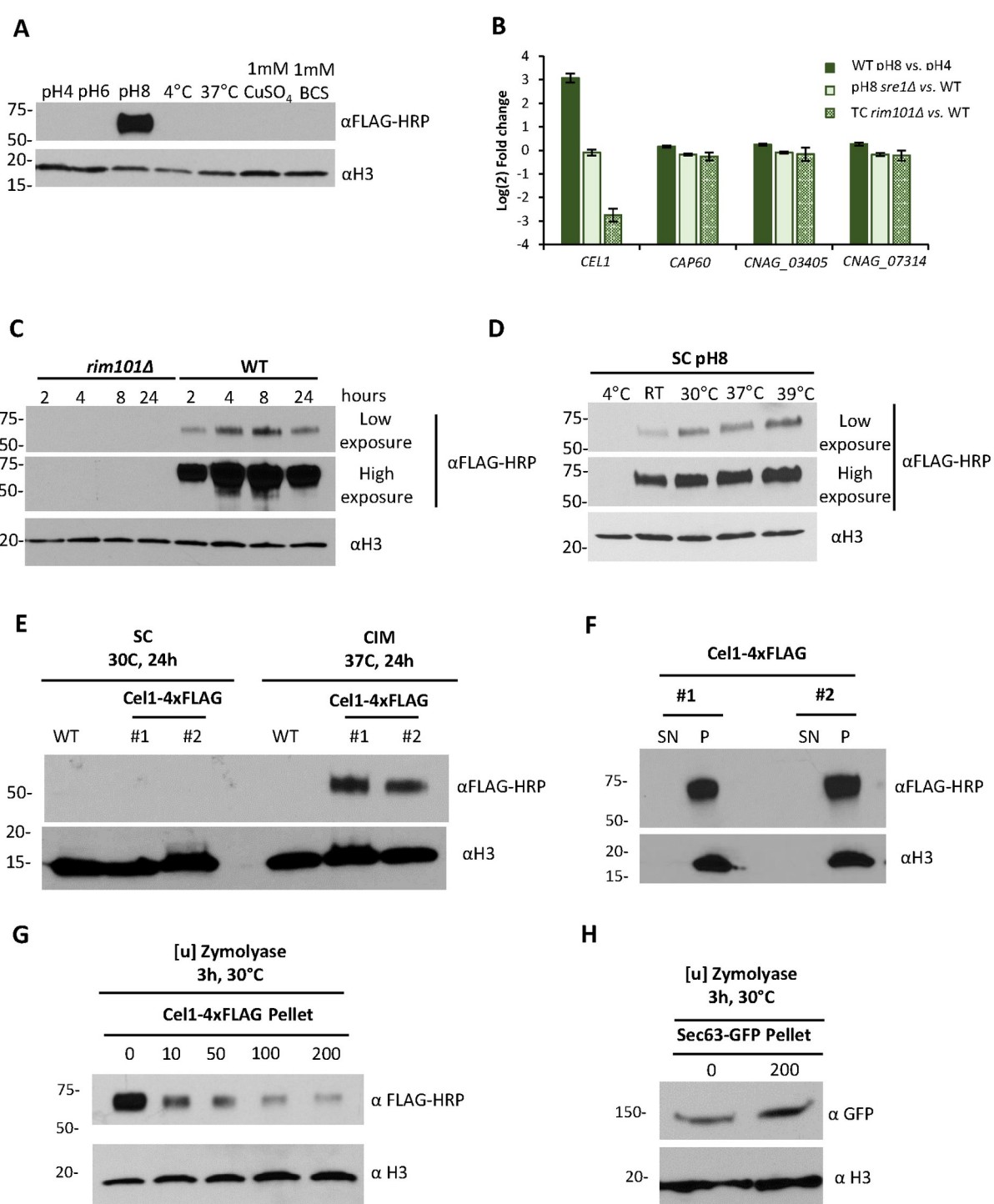

**Fig 2. *CEL1* is induced during alkaline pH and high temperature stress in a Rim101-dependent manner and is associated with the *Cn* cell wall.** (**A**) Cel1-4xFLAG protein expression assessed by western blot. The strain expressing the Cel1-4xFLAG fusion protein was incubated in SC medium (starting $OD_{600}$ of 0.4) for 3h at indicated pH and temperature, or in SC medium supplemented with 1mM $CuSO_4$ or 1mM BCS as indicated. Protein extraction was performed by TCA-based protein extraction. For each condition, 20 μL of crude TCA extract was analyzed by Western blot for Cel1-4xFLAG using the αFLAG-HRP conjugate antibody. The αH3 (anti-histone 3) antibody was used as a loading control. (**B**) Transcript abundance of *Cn CEL1*, *CAP60*, and two other predicted *Cn* LPMOs (CNAG_03405 and CNAG_07314). Previously reported RNASeq data was analyzed to quantify alterations in the expression of these four genes in response to changes in pH, as well as in two mutants impaired for growth at high pH (*sre1Δ* and *rim101Δ* mutant strains): Brown *et al.*, 2020 [29] (WT pH8 vs. pH4 and pH8 *sre1Δ* vs. WT) and Brown *et al.*, 2018 [28] (Tissue culture (TC) *rim101Δ* vs. WT). Transcript data were plotted with

GraphPad Prism and are shown in $\log_2$-fold change. **(C)** Cel1-4xFLAG protein expression analysis by western blot in response to alkaline pH stress in the WT and *rim101Δ* strain. The two strains were incubated at 30°C in SC medium pH 8.15 (starting $OD_{600}$ of 0.4) for indicated times, followed by TCA-based protein extraction. For each condition, 25 μL of crude TCA extract was analyzed by western blot for Cel1-4xFLAG expression using the αFLAG-HRP conjugate and αH3 as loading control. **(D)** Temperature-dependent Cel1-4xFLAG protein expression at alkaline pH. The strain expressing the Cel1-4xFLAG fusion protein was incubated in SC medium pH 8.15 (starting $OD_{600}$ of 0.4) for 3h at indicated temperatures, followed by TCA-based protein extraction. For each condition, 25 μL of crude TCA extract was analyzed by western blot for Cel1-4xFLAG expression using the αFLAG-HRP conjugate and αH3 as loading control. **(E)** Cel1-4xFLAG protein expression in tissue culture medium. Two independent strains expressing the Cel1-4xFLAG fusion protein and the WT strain were incubated in either SC medium at 30°C or in $CO_2$-independent tissue culture medium (CIM) at 37°C for 24 h (starting $OD_{600}$ of 0.1), followed by TCA-based protein extraction. For each condition, 20 μL of crude TCA extract was analyzed by western blot for Cel1-4xFLAG expression using the αFLAG-HRP conjugate and αH3 as loading control. **(F)** Western blot analysis of Cel1-4xFLAG secretion pattern. Two independent strains expressing the Cel1-4xFLAG protein were incubated in CIM (starting $OD_{600}$ of 0.1) for 24h. Cell pellets (P) and supernatant (SN) were harvested, and the supernatant was filtered (0.2μ filter) to remove residual cells. Total protein from each sample was extracted using the TCA-based protein extraction method. For each condition, 20 μL of crude TCA extract was analyzed by western blot for Cel1-4xFLAG using the αFLAG-HRP conjugate and αH3 as loading control. **(G-H)** Western blot analysis of Cel1-4xFLAG (G) and Sec63-GFP (H) protein levels upon cell wall degradation. Cel1-4xFLAG expression was induced as described in (F). Sec63-GFP was grown overnight in YPD at 30°C and then cultivated for 3h in SC+250μM $CuSO_4$ at 30°C. Cell wall degradation was performed using increasing amounts of Zymolyase (10u, 50 u, 100u, 200u). As control for Cel1-4xFLAG and Sec63-GFP protein levels in the intact cell wall, the assay was performed without the addition of Zymolyase (0u sample). Total protein from each sample was extracted using the TCA-based protein extraction method. For each condition, 20 μL of crude TCA extract was analyzed by western blot for Cel1-4xFLAG using the αFLAG-HRP conjugate (G) and for Sec63-GFP (H) using the αGFP antibody. Histone3 levels were determined using the αH3 antibody.

the *CEL1* gene (*cel1Δ^C* complemented strain) to ensure that any observed mutant phenotypic alterations were in fact due to the *cel1Δ* mutation. We first assessed the relative virulence of the *cel1Δ* mutant in a murine inhalational model of cryptococcal infection (Fig 3A). This model mimics the natural route of infection acquisition and typically results in systemic dissemination and host death. C57BL/6 mice were inoculated by nasal inhalation with $10^5$ fungal cells of either the wildtype, *cel1Δ* mutant, or *cel1Δ^C* complemented strain. The wild-type and complemented strains demonstrated a similar level of virulence in this model, with 50% of the infected animals dying from the infection after 21–24 days. In contrast, all mice infected with the *cel1Δ* mutant strain survived 8-weeks after inoculation, with none exhibiting symptoms associated with infection (weight loss, decreased activity, neurological symptoms). Moreover, no viable *cel1Δ* mutant cells were recovered by culturing homogenates of infected mouse lungs at the end of this experiment (S3 Fig), indicating that the animals likely sterilized the infection with this highly attenuated strain.

To further analyze the survival rate of the *cel1Δ* strain *in vivo*, we performed quantitative cultures of infected mouse lungs at 3, 7, and 14 days after infection (Fig 3B). After 3 days of infection, the *cel1Δ* strain showed a significant reduction in fungal colony-forming units (CFUs) in the infected lungs. In contrast to WT, the *cel1Δ* mutant CFU analysis suggested a reduction in total viable cells compared to the initial inoculum, demonstrating fungal killing *in vivo* during the initial stages of infection (*cel1Δ* 3 days post infection ~$3x10^2$ colonies/g lung; initial inoculum $10^5$ CFUs). This finding is consistent with histological analysis of infected mouse lungs in which numerous WT cells are visible in the lung parenchyma 3 days post-infection, and only rare fungal cells are observed in the lungs of *cel1Δ* infected animals at this same time point (Fig 3C). At 7 and 14 days post-infection, the lung fungal burden increased exponentially in the animals infected with the WT strain. In contrast, the *cel1Δ* mutant displayed very low survival rates during these early time points. Together with the apparent sterilization of *cel1Δ*-infected lungs at later time points (S3B Fig), these results suggest that the *cel1Δ* strain is unable to establish a productive lung infection, fails to proliferate in the host lung environment, and is eventually cleared over time.

We also tested the ability of these strains to cause a lethal infection in larvae of the Greater wax moth (*Galleria mellonella*). This invertebrate infection model assesses early innate immune responses in an invertebrate host to a fungal challenge, and the rate of host death in

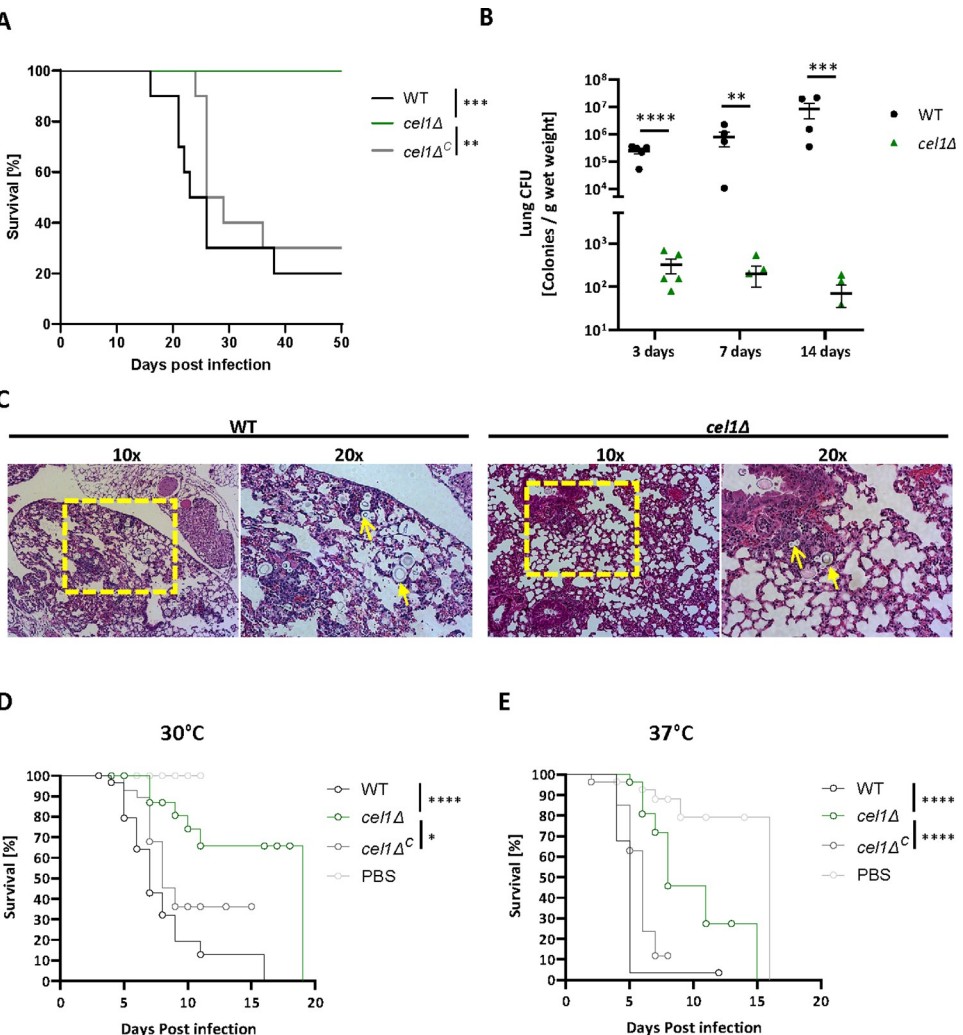

**Fig 3. *cel1Δ* shows decreased virulence in two different models of cryptococcosis. (A)** Survival in murine inhalation model of cryptococcosis. Female C57BL/6 mice were infected by inhalation with $10^5$ cells of the indicated strains (10 mice per strain). Mice were monitored twice daily and sacrificed when 15% weight loss was achieved or when exhibiting symptoms linked to imminent mortality. Survival data were plotted, and a log-rank test as statistical analysis was performed using GraphPad prism. **(B)** Lung colonization at early stages of murine infection. Female C57BL/6 mice (5 mice per strain) were infected with indicated strains as described in (A). At timepoints indicated mice were sacrificed, the lungs were harvested, and lung lysates were assessed by quantitative culture, normalized by lung wet weight. Data were plotted and an unpaired t-test was performed using GraphPad Prism. **(C)** Lungs of female C57BL/6 mice infected with $10^5$ WT or *cel1Δ* cells sacrificed at 3 d post infection were harvested for histopathological analyses. Hematoxylin and eosin staining was used to visualize microscopic lung pathology. Fungal cells are indicated with yellow arrowheads. The area used for 20x magnification is marked by a yellow box. **(D-E)** Lethal infection study in larvae of the Greater wax moth *G. mellonella*. Larvae were infected by injection of $10^7$ cells of indicated strains (30 larvae per strain). Infected larvae were incubated at 30˚C (D) or 37˚C (E) and monitored daily for viability (movement and melanin production) and pupae formation. Log-rank test was performed using GraphPad prism. Pupae formation events are marked as an "0" within the survival curve.

this system is highly correlated with virulence in mammalian infection models [32]. However, since these larvae can survive at 30˚C and 37˚C, this insect infection allows us to separate alterations of virulence from simple impairments in thermotolerance. Thirty larvae for each strain and each incubation temperature were infected with $10^7$ cells of the WT, *cel1Δ* mutant, or *cel1Δ^C* complemented strain. Larvae infected with either the WT or complemented strains

displayed similar rates of host death at both temperatures (median survival 7–8 days at 30˚C, and 5–6 days at 37˚C, Fig 3D–3E). In contrast, there was a significant increase in host survival at both temperatures after infection with the *cel1Δ* mutant: 30% mortality at 30˚C after 20 days of observation, and a median survival of 8 days at the more stressful host temperature of 37˚C. Therefore, *Cn*Cel1 is required for *Cn* pathogenesis in two independent infection models in a manner that suggests that this protein is expressed and functioning *in vivo* at both 30˚C and 37˚C.

In plant pathogenic fungal species, LPMO's have been proposed to promote the establishment of infection on the plant surface by degrading structural polysaccharides such as ligno-cellulose [15,33]. Prior investigations in *C. neoformans* have reported varying degrees of association with live plants, despite frequent isolation of this fungus from decaying plant material [34]. Using previously described methods of *Cryptococcus–Arabidopsis thaliana* co-incubation models [35–38], we observed no differences in plant association between WT, *cel1Δ*, and *cel1Δ*^C strains by scanning electron microscopy of the plant surface (S3C Fig). Moreover, we observed few pathological signs on plants for any of these strains, despite using several means of inoculation (scarification, infiltration, spray, and drop methods) over a 3-week period of observation, or until plant flowering (S3C Fig).

## Loss of *Cn*Cel1 impacts cell wall integrity and homeostasis in the presence of high temperature and alkaline pH stress

To further explore mechanisms for the profound hypovirulence of the *cel1Δ* mutant in animal infections, we quantified alterations in fitness and virulence-associated phenotypes. We first assessed the ability of the *cel1Δ* mutant to adapt to challenges to cell surface integrity, including elevated incubation temperatures and the addition of cell surface stressors to the growth medium at both acidic (YPD pH 4.15 and pH 5.6) and alkaline pH (YPD pH 8.15). The WT, *cel1Δ*, and *cel1Δ*^C complemented strains displayed similar growth rates at 30˚C on YPD medium. In contrast, the *cel1Δ* mutant had a moderate growth impairment at 37˚C, and this thermotolerance defect was enhanced by the cell wall stressors calcofluor white (CFW), caffeine, SDS, and Congo Red (CR) (Fig 4A).

The *cel1Δ* strain displayed mild growth impairment at alkaline pH at 30˚C. However, there was a striking enhancement of this pH-related growth defect at 37˚C (Fig 4A), representing two conditions in which the Cel1-4xFLAG protein is highly induced *in vitro*. We also observed enhanced sensitivity of this strain at pH 8.15 to SDS, CFW, Caffeine and Congo red (Fig 4B). Strikingly, no sensitivity to cell wall stressors was observed on a rich medium (YPD pH 5.6) at 30˚C (Fig 4A and 4B). We further assessed the role of *Cn*Cel1 in *C. neoformans* cell wall adaptations by testing the *Cn* WT, *cel1Δ*, and *cel1Δ*^C complemented strains as well as the chitin synthase 3 mutant strain (*chs3Δ*), a strain lacking cell wall chitosan with a known decrease in cell wall integrity [39], for resistance to *Trichoderma harzianum* lysing enzymes, a mixture of chitinases and glucanases that degrade the complex polysaccharides present in fungal cell walls (S3E Fig). In line with the previous finding, the *cel1Δ* strain displayed an increased susceptibility to cell wall lysing enzymes in presence of alkaline pH and high temperature stress, which further supports a model for *Cn*Cel1 in contributing to cell wall integrity during alkaline and high temperature stress.

Cinnamtannin B1, a proanthocyanin from the bark of cinnamon, has been reported to inhibit LPMO redox activity [40]. Therefore, we aimed to test whether cinnamtannin B1 affects growth of *Cn* specifically in a condition where *CEL1* is expressed (host pH). To do so we performed a minimal inhibitory concentration analysis (MIC) using a cinnamon extract containing cinnamtannin B1 in YNB or YNB buffered to pH 7.4 (YNB-pH 7.4) at 30˚C (S3G–

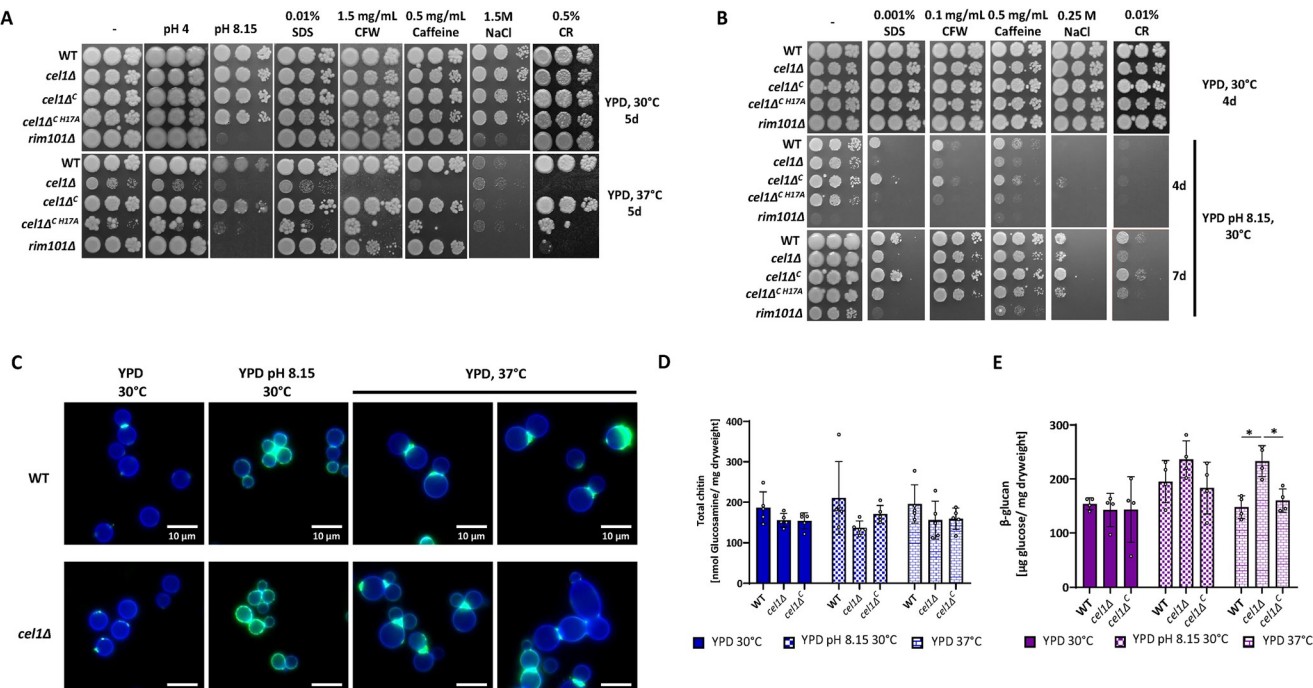

**Fig 4. *Cn*Cel1 affects cell wall integrity under high temperature and alkaline pH stress. (A-B)** Growth analysis in the presence of cell wall/ surface stressors and high temperature (A) or alkaline pH (B) stress. Five-fold serial dilutions of cell suspensions for each strain were incubated on YPD medium (-) or pH-buffered YPD medium at 30°C or 37°C for 4-7d as indicated in the figure. Cell wall stressors were supplemented to the YPD medium as indicated. **(C)** Microscopic analysis of CFW and WGA-Alexa-488 stained cells. The WT and *cel1Δ* strains were incubated overnight in YPD medium and resuspended to $OD_{600}$ 0.1 in indicated medium (either YPD or YPD pH 8.15). Cells were cultivated for 24h hours at indicated temperatures and stained with CFW (total cell wall chitin) and WGA-Alexa488 (exposed cell wall chitin). Shown are representative images of this analysis. The CFW signal (blue channel) and WGA-Alexa 488 (green channel) are presented as merged images. Cell wall quantification of **(D)** total chitin and **(E)** β-glucan. Indicated strains were incubated for 24h in YPD 30°C, YPD pH 8.15 30°C or YPD 37°C. Cell wall material was purified from lyophilized yeasts, and total chitin was quantified using the MBTH-based chitin/chitosan quantification method. β-glucan was quantified from lyophilized yeast cells using the Megazyme yeast β-glucan kit. Data represent the mean +/- SEM of 4 biological replicates. A 2-way ANOVA was performed from log-transformed data using GraphPad Prism.

S3F Fig). While only a slight growth defect was observed in unbuffered YNB (~pH 5.4), strains grown in YNB-pH 7.4 showed growth inhibition in presence of 1.2 µg/mL cinnamon extract (S3G Fig). No difference was observed between the WT and *cel1Δ* strain in growth and inhibition in the different media. The cinnamon extract was dissolved in 50% DMSO; a parallel control experiment with 50% DMSO demonstrated no significant growth inhibition from the solvent alone in YNB or YNB-pH 7.4 (S3F Fig). These results suggest that compounds present in the cinnamon extract, such as cinnamtannin B1, are targeting pathways specifically required for growth during alkaline pH stress, which may include *Cn*Cel1 but also potentially other pH-responsive processes.

To analyze the requirement of an intact copper histidine brace for *Cn*Cel1 function, we transformed the *cel1Δ* mutant with a *cel1* mutant allele with a single nucleotide change resulting in a histidine to alanine substitution at amino acid position 17 (strain *cel1Δ*^*C H17A*). After post-translational protein processing and removal of the signal peptide, this histidine residue is predicted by sequence homology to be the initial amino acid (His1) in the mature protein (S1A Fig), and it is also predicted to be required for copper binding and LPMO enzymatic activity [24]. Validation of the H17A mutation in the transformants was confirmed by PCR amplification and sequencing of the *CEL1* allele from genomic DNA and expression of the *CEL1*-H17A gene allele was validated by q-RT-PCR analysis (S3D Fig). Consistent with the

proposed role of the histidine brace for LPMO function, the mutated Cel1-H17A protein did not complement the growth phenotypes of the *cel1Δ* strain, in contrast to expression of a reintroduced WT *CEL1* allele that resulted in full phenotypic complementation (Fig 4A and 4B).

To determine if the alterations in cell surface stress resistance were associated with changes in cell morphology, we performed microscopic analysis of Calcofluor white (CFW) and Alexa-488 conjugated wheat germ agglutin (WGA) co-stained WT and *cel1Δ* cells after 24h growth at permissive growth condition (YPD, 30˚C), alkaline pH stress (YPD pH 8.15, 30˚C) and high temperature stress (YPD, 37˚C). CFW is a small molecule staining 1,3-β- and 1,4-β-polysaccharides, which include chitin and chitosan [41]. WGA is a lectin that binds to exposed chito-oligomers [42,43]. Therefore, a CFW/WGA co-stain demonstrates both the amount of chitin present in the cell wall as well as its surface exposure, thus providing an indirect image of changes in the fungal cell surface. No difference in CFW staining was observed between the WT and *cel1Δ* strains in any of the analyzed growth conditions, suggesting no major alterations in overall chitin deposition within the *cel1Δ* cell wall. However, the *cel1Δ* strain exhibited a notable change in budding behavior at high temperatures compared to WT, including an increased number of cells with multiple buds and wide bud necks (Figs 4C and S4).

To more quantitatively explore changes in cell wall carbohydrate content, we measured total chitin, chitosan and β-glucan in the WT, *cel1Δ* and *cel1Δ$^C$* strains incubated at permissive growth conditions, alkaline pH stress and high temperature stress for 24h. In line with the previous microscopic findings of CFW-stained cells, no significant changes were measured in total chitin or chitosan in the harvested cell wall material of the *cel1Δ* strain compared to WT (Figs 3G and 4D). Interestingly, we noted increased total β-glucan levels in the *cel1Δ* cell wall compared to WT after high temperature stress, the stress condition in which *cel1Δ* cells showed the most striking changes in morphology and budding behavior (Figs 4C, 4E and S4).

## The *cel1Δ* strain displays altered virulence-associated phenotypes

Cell wall remodeling during infection promotes the expression of several virulence-associated phenotypes such as the formation of the antioxidant melanin pigment, the expansion of cell surface capsule, and the morphological transition to titan cells *in vivo* [5,6,30,39]. Mutant strains with defects in cell wall architecture and/or remodeling often demonstrate defects in these phenotypes [30,39,44].

### Melanin

In contrast to the WT and complemented strains, the *cel1Δ* mutant displayed impaired expression of melanin on L-DOPA medium, especially at 37˚C (Fig 5A). *Cn*Cel1 is a copper-containing enzyme, and similar reductions in melanin production are often observed in strains with defects in copper acquisition [4,45]. However, in contrast to copper homeostasis mutants, the melanin defect of the *cel1Δ* strain was not restored by copper supplementation (Fig 5A), suggesting that cellular changes other than alterations of copper availability are responsible for the *cel1Δ* mutant melanin phenotype.

### Capsule and cell surface architecture

The cryptococcal capsule is composed of highly branched polysaccharides covalently attached to α-1,3 glucans in the outer layer of the cell wall [12]. Capsule polysaccharides are constantly produced and secreted, however capsule attachment and expansion at the cell surface are induced during incubation in conditions that mimic the host environment [46].

To analyze capsule attachment and expansion at host mimicking stress conditions, we incubated the WT and *cel1Δ* strains in CIM at 37˚C and assessed the presence of surface capsule by

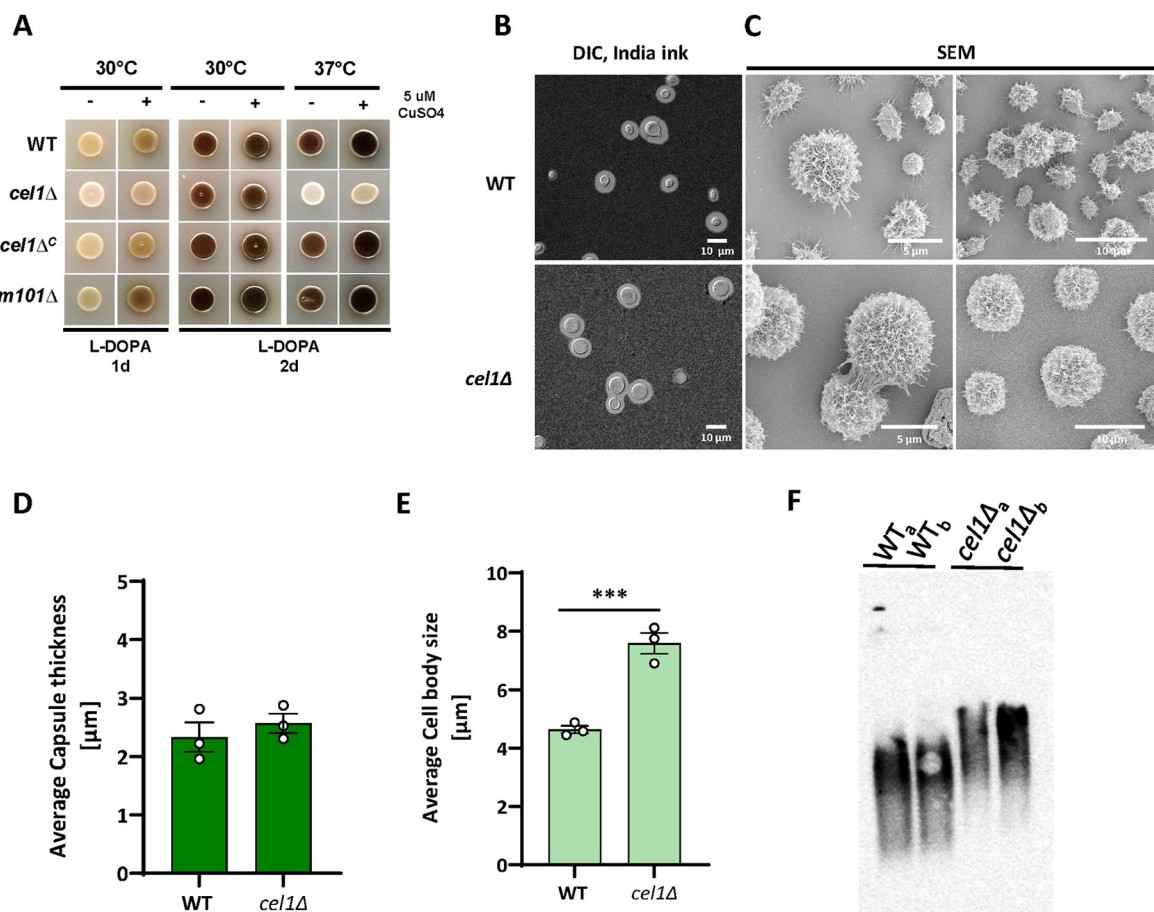

**Fig 5. *Cn*Cel1 affects melanization and capsule architecture. (A)** Melanization in the absence and presence of exogenous copper. Indicated strains were incubated on L-DOPA plates at 30°C or 37°C, with and without supplementation with exogenous CuSO4. Melanin formation was assessed at 1–2 days as indicated. **(B-E)** Surface capsule formation and architecture. The WT and *cel1Δ* mutant were incubated for 3 days in CIM tissue culture medium to induce capsule. Surface capsule characteristics were assessed by (B) India ink counter-staining and (C) SEM. Representative images are shown from 3 independent experiments. Quantification of capsule size (D) and cell body size (E) was performed using ImageJ/Fiji from static images of india ink-stained cells. Mean values from >100 cells per sample (+/- SEM) from 3 biological replicates are shown as bar graph. Data were plotted, and an unpaired t-test was performed using GraphPad Prism. **(F)** Electromobility of exopolysaccharide. The WT and *cel1Δ* mutant were incubated in CIM for 3 days. Exopolysaccharide in the culture supernatant was assessed for relative size and gel motility by agarose gel electrophoresis, transferred to a Nylon membrane, and probed with anti-GXM antibody 18B7. Representative image of multiple replicates is shown.

India ink counter-staining after 3 days of culture. Each strain displayed a uniform pattern of cell surface capsule induction with similar capsule diameters (Fig 5B and 5D). Using SEM to better resolve the capsule and cell surface structure, we observed morphological changes in the *cel1Δ* strain as compared to WT, showing less interconnected and extended capsule fibers and an enlarged cell body (Fig 5C and 5E). To explore potential structural changes in the secreted capsule material, we analyzed the electrophoretic mobility of the capsular exopolysaccharides secreted into the growth medium as an estimate of polymer size and degree of branching complexity [47] (Fig 5F). Both WT and *cel1Δ* cells secreted similar amounts of capsule material. However, there was a slower overall electrophoretic mobility of the capsular exopolysaccharide material secreted by the *cel1Δ* strain compared to WT, suggesting differences in branching and/or composition.

## Titanization and cell cycle progression

During mammalian lung infection, *Cn* undergoes a morphological transition from small (5–6μm) yeast cells to enlarged (>15μm) titan cells. These polyploid and heavily encapsulated cells are resistant to many host stresses and associated with improved survival during the initial stages of infection. Using *in vitro* conditions that induce titan cell transitions, we quantified cell size distribution of WT, *cel1Δ*, and *cel1Δ*^C complemented strains, as well as the titanization-defective *rim101Δ* strain [48] after 24, 48 and 72h of growth at low cell density in titan-inducing medium. As expected, no titan cells were observed in the *rim101Δ* strain at any of the time points (Fig 6A–6D). In contrast, after 24h ~15% of WT and complemented cells were larger than 10 μm, and many had cell diameters greater than 15 μm, consistent with a progressive titan cell transition (Fig 6A). Although 40% of *cel1Δ* cells had an enlarged diameter in these growth conditions, they were more uniform in cell size and failed to produce a significant number of cells larger than 15 μm. By 48-72h of incubation, the WT and complemented strains demonstrated an expected pattern of mixed cell sizes, including very large titans (> 15 μm), normal sized cells, and very small titanides (oval cells < 3 μm likely produced from titan cells) (Fig 6B and 6C) [49]. In contrast, the *cel1Δ* mutant displayed a more uniform cell body size,

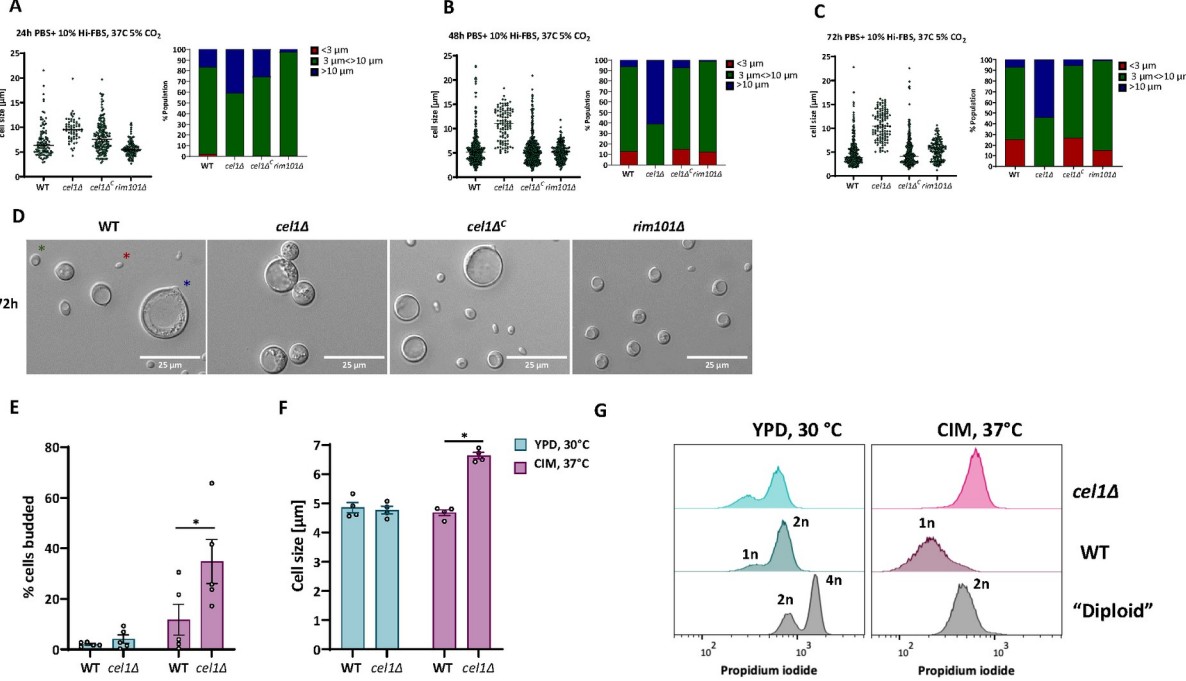

**Fig 6. *cel1Δ* shows cell cycle defects under host like stress conditions. (A-D)** *In vitro* titanization. The indicated strains were incubated overnight in YNB medium, diluted to $OD_{600}$ of 0.001 in titan cell-inducing medium (PBS+10% HI-FBS, 37˚C, 5%$CO_2$) [49], and further incubated for 24h, 48h, and 72h. Cells were analyzed by microscopy and cell diameter measured using Image J/Fiji. **(A-C)** Scatter plots and bar charts of cell size distribution are demonstrated for indicated times. All graphs were generated using GraphPad Prism. **(D)** Representative DIC images of indicated strains after 72h growth in titan cell-inducing conditions. **(E)** Budding index (ratio of budded:total cells) of indicated strains after 18h of conditioning in YPD at 30˚C or CIM at 37˚C. A minimum of 300 cells were assessed per strain and condition. Shown is the mean +/- SEM of 5 biological replicates. Data were plotted using GraphPad Prism. A 2-way ANOVA was performed from log transformed data. **(F)** Average cell size of indicated strains after 18h of conditioning in YPD at 30˚C or CIM at 37˚C. A minimum of 150 cells was measured per strain and condition. Shown is the mean +/- SEM of 4 biological replicates. Data were plotted using GraphPad Prism. A 2-way ANOVA was performed from log transformed data. **(G)** Ploidy analysis in host-mimicking conditions. Indicated strains were inoculated to a starting $OD_{600}$ of 0.1 and incubated for 24h in YPD at 30˚C or CIM at 37˚C. Cells were fixed and stained with propidium iodine. Propidium iodine staining was quantified by Flow Cytometry. Data were analyzed using the FlowJo software. A representative histogram is shown from 3 independent ploidy analyses.

though slightly increased from that during incubation in more permissive growth conditions (Fig 6B–6D). Only a small number of cells larger than 15μm and no titanides were observed among the *cel1Δ* cells.

In previous experiments we observed an enlarged cell body in non-titan-inducing stress condition as well as an aberrant budding behavior when *cel1Δ* cells were grown in presence of high temperature stress. We therefore tested whether the large cell size of the *cel1Δ* mutant observed during stressful growth conditions was due to failed cell cycle progression, rather than titanization. The WT, *cel1Δ*, and complemented strains were incubated in both permissive growth condition (YPD, 30°C) and a host stress-inducing condition (CIM at 37°C), and cell cycle progression was assessed by measuring ploidy, cell morphogenesis, and relative budding index (budded cells/unbudded cells) over time. In permissive growth conditions, no differences were noted between the WT and the *cel1Δ* strains for cell size, morphology, or budding index after 18h of conditioning (Fig 6E). However, during growth in CIM at 37°C, the *cel1Δ* strain exhibited a significant increase in budding index compared to WT. These changes in budding frequency in the *cel1Δ* strain were associated with an increase in the average cell body diameter compared to WT (Fig 6F). Flow cytometry of propidium iodine (PI)-stained cells revealed no differences in DNA content between WT and *cel1Δ* cells when cells were grown under permissive growth conditions for 24h. In both strains, the cell population was composed of cells with 1n and 2n DNA content, as expected for normal cell cycle progression. In a stable diploid strain as a control, the cell population was composed of cells with 2n and 4n DNA content in this condition, as previously reported [50]. However, the *cel1Δ* mutant arrested with a 2n DNA content after prolonged incubation in CIM at 37°C, distinct from the 1n DNA content observed in WT cells (Fig 6G). Together, these data suggest that *Cn*Cel1 is required for efficient cell cycle progression during host-relevant stresses.

## Discussion

Since their discovery, LPMOs have mainly been studied for their role in biomass degradation [14,51]. LPMOs possess a flat surface with an exposed Cu-active site that interacts with crystalline polysaccharide substrates, catalyzing the oxidative cleavage of glycosidic bonds [24]. In this way, LPMOs create nicks in flat biopolymer sheets to allow additional hydrolyzing enzymes to further saccharify carbohydrate structures [52,53]. Even though LPMOs were first described as enzymes involved in dead biomass recycling, they have been functionally linked to new roles such as cell morphogenesis. Additionally, for plant- and insect-targeting microbes, LPMOs are important effectors during commensalism and pathogenic interactions [23,54–58]. In these previous studies, LPMOs were shown to be secreted together with other CAZymes for degradation of the host cell wall to facilitate microbial invasion of the host.

In contrast to plant pathogens, studies about the role of LPMOs in human pathogenic microbes are limited to bacterial LPMOs. Recent investigations identified chitin-active LPMOs from *Vibrio cholerae*, *Listeria monocytogenes* and *Pseudomonas aeruginosa* to be required for full virulence [22,58,59]. However, the exact function and substrates of those bacterial LPMOs during infection have yet to be elucidated.

In our study we have uncovered a fundamental role of the *Cn*Cel1 protein in stress adaption and virulence of the human fungal pathogen *C. neoformans*. The *Cn*Cel1 protein phylogenetically groups with AA9 cellulose-active LPMOs, sharing the aromatic amino acids known to be involved in AA9 LPMO substrate recognition, the copper binding histidines, as well as the secondary sphere coordinating tyrosine [60,61]. Consequentially, the expression of an N-terminal histidine to alanine *Cn*Cel1 variant (*cel1Δ*$^{C\ H17A}$), does not complement the stress phenotypes of the *cel1Δ* mutant strain, demonstrating that an intact active site is essential for *Cn*Cel1

function. In line with this observation, our *in vitro* enzyme activity assay also confirms that *Cn*Cel1 is redox-active and copper dependent. Initial substrate screening is ongoing and so far testing on the most common substrates (avicel, PASC, β-glucan, xylan, arabinoxylan, chitin, chitosan, and glucomannan) has all been negative (results not shown). Further detailed biochemical characterization is required to identify the *in vivo* substrate. Interestingly, compared to *Ta*AA9A, *Cn*Cel1 has several additional aromatic residues at the substrate binding site, which could broaden or alter the substrate binding specificity. Nevertheless, our initial biochemical characterization of *Cn*Cel1 *in vitro* activity together with our fungal physiological studies have begun to elucidate the importance of this enzyme in intrinsic fungal cell processes.

A *cel1Δ* mutation results in complete loss of virulence and rapid microbial clearance from the host. This is the first report of a fungal-specific LPMO as a virulence determinant, suggesting that Cel1-like LPMOs might be involved in virulence in other clinically relevant fungi. Phenotypic analysis of the *cel1Δ* mutant suggests that this protein is involved in cell surface remodeling and homeostasis. Compared to WT, this strain is more susceptible to cell wall stresses, especially under conditions in which the *Cn*Cel1 protein is most highly expressed. Moreover, the *cel1Δ* mutant is more susceptible to cell wall lysing enzymes, suggesting an impairment in cell wall structure, adaptation, and integrity. Enzymatic digestion of the cell wall results in a dose-dependent loss of recovery of the Cel1 protein from the fungal cell, suggesting that the *Cn*Cel1 protein is associated with structures within the *Cn* cell wall.

A recent study similarly identified a bacterial LPMO involved in facilitating cell wall remodeling [23]. Like *Cn*Cel1, the LpmP protein in *Streptomyces coelicolor* (*S. coelicolor*) is retained in the bacterial cell wall, rather than being secreted into the environment. At this site, LmpP likely functions with other cell wall modifying enzymes at localized spots of cell wall degradation to facilitate the passage of glycan through the inner layer of the bacterial cell wall. Also similar to *Cn*Cel1, loss of function of the bacterial LPMO results in altered susceptibility to cell wall degrading enzymes. Although not yet experimentally proven, the localization of this LPMO within the microbial cell wall might be due to interaction with its target substrate. Together, these data from two different biological kingdoms suggests that LPMOs might serve a conserved function in microorganisms in the active remodeling of cell wall polysaccharides.

A recent report of an LPMO from the filamentous fungus *Neurospora crassa* demonstrated that these proteins might have physiological functions in fungi beyond their enzymatic activity. The CWR-1 and CWR-2 proteins were shown to be required for recognizing strains of different haplotypes that were compatible for filament fusion. Interestingly, and in contrast to our findings in *Cn*Cel1, this activity was independent of its catalytic function as mutation of the copper-binding residues did not affect this aspect of protein function [62].

## *Cn*Cel1 is induced in host-relevant conditions

Several prior transcriptomic studies have shown induction of *Cn CEL1* during various stages of pathogenesis. It is highly expressed during lung infections of female C57BL/6 mice [63], in the CSF of patients with cryptococcal meningitis [21,64], and during phagocytosis by macrophages and amebae [64,65]. *Cn CEL1* transcript was not detected in the lungs of infected Cynomolgus monkeys [63]. *In vitro*, *CEL1* transcript abundance increased in response to osmotic and oxidative stress: Mutations in the stress response regulator genes *HOG1* and *SSK1* did not abolish *CEL1* expression in either condition, but these mutations blunted *CEL1* transcriptional induction by ROS [66]. *CEL1* transcriptional induction appears to be independent of cAMP signaling: mutations in components of this signaling pathway (Gpa1, Aca1, Cac1, Pka1, Pka2) did not change *CEL1* transcript abundance. In contrast, *CEL1* transcript levels were reduced by a *ras1* mutation [67].

Here we have demonstrated that *Cn*Cel1 protein expression is specifically dependent on host physiological temperatures and pH. Moreover, induction of *CEL1* gene transcript abundance at alkaline pH requires intact function of the Rim signal transduction pathway. This fungal-specific signaling cascade is first activated by alkaline pH-induced changes in plasma membrane lipid bilayer asymmetry [28]. Activation of this pathway eventually results in the proteolytic cleavage and nuclear translocation of the Rim101 transcription factor, directing focused transcriptional changes that promote survival at higher pH. *Cn RIM101* is among the most highly induced genes in the setting of an experimental infection [68]. Moreover, many of the Rim101 target genes are involved in cell wall modifications, suggesting that dynamic changes in this structure are required for microbial growth at the pH of the infected host [68,69]. The differences in phenotypes between the *cel1Δ* and *rim101Δ* strains during infection and *in vitro* stresses confirm prior transcriptional analysis that *CEL1* is only one of several cell wall modifying enzymes regulated by the Rim101 pathway in response to changing environmental conditions [69].

## Proposed sites of enzymatic activity

Cell wall polysaccharides undergo intermolecular cross-linking to promote cell integrity. Fungal chitin, chitosan, and glucans are made by cell membrane-associated synthases and deposited in the extracellular space. Transglycosylases create covalent cross-linkages, such as the CRH protein family that promote chitin-glucan linkages in *S. cerevisiae* [8]. These emerging models, as well as new cell wall imaging modalities, indicate that fungal cell wall polysaccharides do not exist as simple linear chains, but rather in highly complex three-dimensional structures [70,71]. With their activity to nick planar sheets of complex polysaccharides, LPMOs might allow specific hydrolases to act in less accessible sites to promote cell wall modification and adaptation. We did not observe significant changes in chitin or chitosan content in the *cel1Δ* mutant cell wall. However, we did note increased levels of β-glucan in this strain. Perturbations in fungal cell walls can result in compensatory changes in other cell wall polysaccharides [72–74]. However, it is also possible that *Cn*Cel1 might facilitate β-glucan modifications during stressful growth conditions.

Rapid fungal cell wall alterations are required for efficient cell cycle progression, attachment of the carbohydrate capsule and incorporation of the pigment melanin as well as more dramatic morphological transitions such as *Cn* titan cell formation, and cell budding to promote extrapulmonary dispersal [5,13,39,44,75–77]. Our phenotypic analysis demonstrated that loss of *Cn*Cel1 affects phenotypes that are dependent on cell wall remodeling in the presence of host stresses such as high temperature and alkaline pH.

One of the hallmark phenotypes of the *cel1Δ* mutant was the formation of enlarged cells when cultivated in different media and at different cell densities. Further analysis demonstrated a budding defect and a 2n arrest when cultivated at host pH and temperatures. Interestingly, these enlarged diploid cells not only fail to progress in the cell cycle, but they also fail to produce fully evolved titan cells under inducing conditions. Recent studies by Altamirano *et al.* reported the role of the cyclin Cln1 and the cell cycle in response to *in vivo* stresses and titanization [78]. In this study, they showed that *Cn* undergoes a variant cell cycle when cultivated in presence of stress, referred to as the stress cycle, harboring a new population of non-budding cells arrested in the G2 phase. These studies also described the role of Cln1 for controlling titan cell formation from the G2 arrest state. Based upon these findings we propose that the *cel1Δ* mutant is arresting in G2 and does not efficiently progress further into the cell cycle, nor is it able to transition towards the formation of titan cells, thus creating a significant population of larger, diploid cells, when cultured under host stress conditions.

The mechanism of the altered cell cycle progression in the *cel1Δ* strain has yet to be fully elucidated. One possibility could be that the effects of the *cel1* mutation on cell wall integrity during stress are sensed by cell surface sensors and thus prevent further progression out of the G2 phase. This model is consistent with the hypothesis that *Cn*Cel1 is involved in facilitating the adaptive changes of the fungal cell wall required for budding initiation and thus for efficient cell cycle progression by providing plasticity to the cell wall structure. Similarly, others have recently emphasized the importance of fungal cell wall structural and compositional flexibility to adapt to host-derived stresses for morphological responsiveness and immunological shielding [2]. This model in which an inflexible fungal cell wall is unable to allow efficient morphological transitions during infection would perhaps explain the defects in cell budding and morphogenesis during stress observed in the *Cn cel1Δ* mutant with impaired cell wall modifying activity.

## LPMOs are potentially targetable enzymes

Recent investigations examined plant extracts for natural products that inhibit the enzymatic activity of LPMOs [40]. These studies were based upon the observation that plant fungal pathogens often encode a greater number of LPMOs compared to plant symbiotes [33,79,80]. Moreover, expression of LPMO genes in the pathogenic *Colletotrichum* species undergoes dramatic induction during the transition from saprobic to pathogenic life stages [81]. Therefore, anti-LPMO activity might be an important aspect of plant defenses against microbial invasion.

The polyphenol cinnamtannin B1 was identified as the active species within a methanol extract of cinnamon that inhibited fungal AA9A LPMO activity from *Lentinus similis* [40]. Variable degrees of inhibition of other cellulose-active LPMOs by cinnamtannin B1 suggest that natural enzyme inhibitors may be quite specific in their function. In our study we tested the inhibitory effects of cinnamon extract, from which cinnamtannin B1 was purified, on *Cn* growth at host pH, a stress condition inducing *CEL1* expression. We showed a strong and pH-specific inhibitory effect of cinnamon extract on growth of WT and *cel1Δ* cells. Interestingly, incubation with cinnamon extract caused a greater inhibitory growth effect in the WT than a loss of function mutant strain of the *CEL1* gene when grown at YNB-pH 7.4. This finding suggests that the observed inhibitory effects of the cinnamon extract cannot solely be explained by the inhibition of *Cn*Cel1 activity. Cinnamon extracts have previously been shown to contain antimicrobial compounds [82] and other inhibitors of carbohydrate active enzymes [83], which could be targeting other *Cn* pathways required for the adaption to the host pH. In conclusion, these results do not prove the inhibition of *Cn*Cel1 activity by the cinnamon extract but do prompt for future analysis of cinnamtannin B1 against the *Cn*Cel1 enzyme and *Cn* growth as well as investigations to identify other *Cn* pathways affected by compounds present in the cinnamon extract. This ongoing investigation might identify new fungal specific compound which could serve as lead compound for further antifungal development.

## Material and methods

### Ethics statement

All animal experiments in this manuscript were approved by the Duke University Institutional Animal Care and Use Committee (IACUC) (protocol #A102-20-05). Mice were handled according to Duke IACUC guidelines.

### Chemicals

All chemicals, unless otherwise stated, were purchased from Merck/Sigma Aldrich (Saint Louise, USA) in the highest available purity grade.

## Strains, media and growth conditions

*Cn* strains used in this study are shown in **S1 Table.** All strains were generated in the *C. neoformans var. grubii H99* background. For strain creation, DNA was introduced into *C. neoformans* by biolistic transformation [84]. Yeast extract (2%)-peptone (1%)-dextrose (2%) (YPD) medium supplemented with 2% agar and 100 μg ml-1 of nourseothricin (NAT), 200 μg ml-1 of neomycin (G418) or 200 μg ml-1 of hygromycin B (HYG) was used for colony selection after biolistic transformation. Cloning strategies as well as plasmids and oligos used for creation of *Cryptococcus* transformation constructs are described in **S2–S3 Tables.** Transformants were screened by PCR and Southern blot for intended mutations. Complementation strains were verified by PCR analysis and *CEL1* transcript analysis. The presence of Cel1 H17A mutation was confirmed by sequencing. Cel1-4xFLAG and Cel1-GFP expression among relevant transformants was confirmed by western. To create a strain expressing Cel1-4xFLAG in the *rim101Δ* mutant background, the Cel1-4xFLAG WT strain (*CEL1p-CEL1-4xFLAG::NAT*, MATα) was genetically crossed to a rim101::NAT MAT a in a mating assay; strains were co-cultured on MS-Medium for 7 days [35]. Spores were isolated by microdissection and *rim101:: NAT CEL1p-CEL1-4xFLAG::NAT* recombinant was identified by nourseothricin (NAT) marker, PCR validation of the *CEL1p-CEL1-4xFLAG* locus and non-growth on alkaline pH media (*rim101Δ* phenotype).

Strains were cultivated in either synthetic complete (SC) medium (MP Biomedicals) or YPD medium at 30˚C. For buffering media, the indicated medium was supplemented with 150 mM HEPES and the pH was set to the indicated pH by adding 6N NaOH or 37% HCl. To mimic *in vitro* host conditions, cells were grown in $CO_2$-independent medium (CIM, Gibco) at 37˚C.

To analyze cell wall associated phenotypes, caffeine, NaCl, SDS, Congo red, Calcofluor White or lysing enzymes from *Trichoderma harzianum* were added to YPD or YPD pH 8.15 medium in indicated concentrations. For growth phenotype analysis on solid medium plates, a 6-fold serial dilution, starting at $OD_{600}$ 0.25, of strains was spotted and incubated for indicated time and temperature. For assessment of melanization, overnight cultures in YPD were washed once in PBS and resuspended in PBS to $OD_{600}$ 2.5. Next, 5 to 10 μL of the resuspended culture were spotted onto L-3,4-dihydroxyphenylalanine (L-DOPA) media (7.6 mM L-asparagine monohydrate, 5.6 mM glucose, 22 mM KH2PO4, 1 mM MgSO4.7H2O, 0.5 mM L-DOPA, 0.3 μM thiamine-HCl, 20 nM biotin, pH 5.6). L-DOPA plates were incubated at 30˚C or 37˚C for 2 days. To induce capsule, strains were incubated in CIM for 72 hours with shaking at 37˚C, followed by staining with India Ink or fixation for scanning electron microscopy (SEM). To induce titanization, cells were precultured overnight in YNB at 30˚C. Then, cells were washed 1x with PBS and transferred to 10 mL PBS+10% HI-FBS in 6 well tissue culture plates. For low density growth (Titan induction) cells were inoculated to $OD_{600}$ of 0.001. 6 well plates were incubated for up to 72h in a tissue culture incubator at 37˚C and 5% $CO_2$[49].

## Phylogenetic sequence analysis

Sequences of characterized LPMO proteins (Uniprot number or Protein Data Bank (PDB) code provided in the figure) were aligned using the T-COFFEE package [85]. The sequences were trimmed for signal peptides to His1 in the histidine brace. Phylogeny was inferred using PhyML software [86] and the Whelan and Goldman (WAG) amino acid substitution model [87]. Branch support was calculated by 100 bootstrap repetitions (value displayed on tree). The tree was visualized with Interactive Tree Of Life (iTOL) software [88].

### Structure based alignment

The sequence of *Cn*Cel1, without the predicted signal peptide, was aligned with the AA9 sequences from the phylogenetic analysis; *Thermothelomyces thermophilus* (TtAA9E, PDB 3EII), *Neurospora crassa* (NcAA9D, PDB 4EIR), *Phanerochaete chrysosporium* (PcAA9D, PDB 4B5Q), *Thermoascus aurantiacus* (TaAA9A, PDB 2YET), *Lentinus similis* (LsAA9A, PDB 5ACF). For alignment, the EXPRESSO of T-COFFEE package was used [89]. The final alignment was prepared using the ESPript 3.0 web server [90], for secondary structure prediction the structure of *Thermoascus aurantiacus* (PDB 2YET) was selected.

### *In silico* protein modelling

To model the *Cn*Cel1 protein, the sequence was submitted to the phyre2 web server [91]. The AA9 LPMO from *Thermoascus aurantiacus* (TaAA9A) was one of the top hits in the alignment (confidence 100.0, coverage of 70%, % i.d. 29) and was used to construct the model of *Cn*Cel1, with the restriction that His1 should not be methylated in order to have His1 from *Cn*Cel1 in the model. The *Cn*Cel1 model was then aligned with the crystal structure of TaAA9A (PDB: 2YET) in PyMOL to visualize the predicted structure as well as the expected copper binding histidine brace. The alphafold2 [92,93] model of *Cn*Cel1 was downloaded from Uniprot and compared with phyre2 model in PyMOL.

### Recombinant expression of the catalytic domain of *Cn*Cel1 in *Pichia pastoris*

Protein expression was performed using the in-house 3PE Platform (*Pichia Pastoris* Protein Express; www.platform3pe.com/). The gene sequence encoding Cel1$_{cat}$ was synthesized after codon optimization for expression in *P. pastoris* (GenScript). The region corresponding to the native signal sequence was kept and the C-terminal disordered region was truncated at G264, hence forward called Cel1$_{cat}$. The synthesized gene was inserted into the pPICZαA vector (Invitrogen) using BstBI and XbaI restriction sites. Transformation of competent *P. pastoris* SuperMan5 was performed by electroporation with PmeI-linearized pPICZαA recombinant plasmids and zeocin-resistant *P. pastoris* transformants were screened for protein production as described [94]. The best-producing transformants were grown in 2 l of BMGY medium in shaken flasks at 30°C in an orbital shaker (200 rpm) to an optical density at 600 nm between 2 and 6. Cells were then transferred to 400 ml of BMMY medium containing 1 ml l$^{-1}$ of PTM$_4$ salts at 20°C in an orbital shaker (200 rpm) for 3 days, with supplementation of 3% (vol/vol) methanol every day. *P. pastoris* strain X33 and the pPICZαA vector are components of the *P. pastoris* Easy Select Expression System (Invitrogen). All media and protocols are described in the manufacturer's manual (Invitrogen). The fermentation supernatant was collected and frozen until chromatographic procedures were started.

### Cel1$_{cat}$ purification

The purification protocol is divided into four steps with three different chromatographic procedures. First, the collected supernatant was buffer exchanged into 25 mM Tris pH 8 using a 200 mL column packed with Sephadex 25g (medium) resin. Second, the exchanged supernatant was applied to a 10 mL Q Sepharose column (GE Healthcare). The protein was eluted by a 0–500 mM NaCl, over 10 CV, gradient. Protein presence was verified using SDS-Page analysis. Third, relevant fractions where pooled and concentrated using 4000g for 35 minutes with 10 kDa MWCO PES spin concentrators (Cytiva). The concentrate was loaded onto a 16/600

Superdex-75 pg column equilibrated with 20 mM MES with 150 mM NaCl at pH 6.6. Protein eluted as a monodisperse peak, as confirmed by SDS-PAGE.

Prior to quantitative amino acid analysis Cel1$_{cat}$ was de-glycosylated using EndoH (Roche Diagnostics GmbH). The de-glycosylation was carried out in 20 mM MES, 150 mM NaCl pH 6.6 using an estimated 10 μl / 1mg Cel1$_{cat}$ (determined by A280). The mixture was incubated at 25˚ C overnight with gentle mixing. Quantitative amino acid compositional analysis was used for accurate determination of the protein concentration in the final Cel1$_{cat}$ batch [95].

## Enzyme activity assays

Cel1$_{cat}$ was Cu-loaded using CuCl$_2$ at 4˚C in a 4:3 sub-stoichiometric ratio. Cu(II)-Cel1$_{cat}$ samples were mixed with 0.125% fluorescein in 75 mM Citrate-Phosphate buffer pH 7.4 with either 100 μM dehydroascorbic acid and/or 100 μM peroxide. The fluorescence of the samples was recorded at 528 nm with excitation at 448 nm every minute for 30 minutes using a Biotek Synergy H1 plate reader.

## RNA extraction and transcript analysis

RNA extraction and transcript analysis was perfomed as described in [4]. Briefly, overnight cultures of indicated strains (YPD, 30˚C) were harvested, washed once with PBS and *CEL1* expression was induced for 3h in SC pH 8.15 at 30˚C. Then cells were harvested, washed once with PBS and lyophilized. RNA was extracted from lyophilized yeast using the RNAesy Plant kit (Qiagen), with on column degradation of DNA using the RNAse free DNAse kit (Qiagen). cDNA synthesis was performed with Iscript (Biorad) using 1μg total RNA. qRT-PCR was perfomed using the PowerUP SYBR Green Master mix (applied biosystems) per protocol instruction and analyzed on a QuantStudio 6 Flex (Applied Biosystems). Oligos used for qRT-PCR analysis are shown in S3 Table. CT values were determined using the QuantStudio 6 Flex software. The houskeeping gene GAPDH was used to normalize gene expression values. Expression fold changes were determined by the ΔΔCT method.

## Protein extraction and immunoblotting

TCA-precipitated protein extracts of *C. neoformans* were used to analyze Cel1-4xFLAG. For all figures, overnight cultures of *C. neoformans* cells were diluted to an OD$_{600}$ of 0.3 in 5 ml of indicated medium and incubated at indicated temperatures and times. To analyze Cel1-4xFLAG induction in response to high and low copper stress, 1mM CuSO$_4$ (high copper) or 1 mM Bathocuproinedisulfonic acid (BCS, low copper) was supplemented to SC medium. For analysis of Cel1-4xFLAG secretion, Cel1-4xFLAG expression was induced for 24h in CIM at 37˚C. The expression culture was centrifuged, and the supernatant was filtered and transferred into a new reaction tube prior to TCA extraction. TCA extraction was performed for both- Pellet and filtered supernatant.

For TCA extraction, cold TCA was added to a final concentration of 10%, cells were pelleted, washed with 20% cold TCA and resuspended in 100 μL 12.5% cold TCA. Resuspended cells were transferred to a 1.5 mL O-ring reaction tube. Glass beads were added, and cells were lysed in a bead beater (Bead Ruptor 12, Omni International) 3 times 60 sec ON and 60 seconds OFF at 4˚C. Cells were transferred into a 1.5 mL reaction tube and centrifuged at 11000x g for 2 mins at 4˚C. The pellet was washed once with 500 mL ice cold acetone. Residual acetone was removed by speed vac (1- 2min). Pellets were resuspended in alkylating buffer (0.1 M Tris pH 8.3, 1mM EDTA, 1% SDS) and incubated for 10 mins at 65˚C. Crude TCA extracts were used in all experiments. Prior to analysis, indicated amounts of the crude TCA extract were treated with SDS-Loading buffer (Biorad) and boiled for 10 mins at 95˚C. TCA protein extracts and supernatants were analyzed by immunoblotting with a monoclonal ANTI-FLAG

M2-Peroxidase with the dilution of 1:5000 (Sigma, αFLAG-HRP), rabbit anti-H3 at 1:1000 dilution (D1H2, polyclonal, Cell Signaling, αH3) and anti-rabbit-HRP in a dilution of 1:4000 (GE Healthcare), mouse anti-CDC2 at 1:1000 dilution (αPSTAIR, abcam). For all primary mouse antibodies an anti-mouse-HRP in a dilution of 1:4000 was used (GE Healthcare). All blots were blocked with TBS-T 5% Milk and washed 3x for 15 minutes with TBS-T after each antibody incubation. Proteins were detected by enhanced chemiluminescence using Super-Signal West Pico (Thermofisher).

## Zymolyase based digestion of the fungal cell wall

Prior to cell wall digestion, Cel1-4xFLAG expression was induced for 24h in CIM at 37°C. The Sec63-GFP expression strain was grown overnight in YPD at 30°C, followed by a 3h growth in SC+250 μM CuSO$_4$ at 30°C (starting OD 0.4). Cells were harvested (3900 rpm, 5 min) and washed once with 0.1 M Tris/HCl pH9. Cells were then resuspended in DTT-buffer (0.1 M Tris/HCl, 10 mM DTT, pH 9.0) and incubated for 20 min at 30°C and 70 rpm. After DTT incubation, cells were washed with Spheroplast buffer (0.1 M Tris, 1.1 M Sorbitol, pH 7.4) once. Cells were resuspended in spheroplast buffer and aliquoted into 5 mL aliquots. To start cell wall digestion Zymolyase yeast lytic enzyme (ZymoResearch) was added to the cells (concentration range: 0 to 200 units), and cells were incubated for 3h at 30°C and 70 rpm. Then, cells were harvested (2200x g 8 min), the supernatant removed and the pellet resuspended in PBS and a TCA based protein extraction was performed as described earlier. Cel1-4xFLAG protein levels were analyzed via western blotting as described earlier. Sec63-GFP protein levels were analyzed using a mouse anti-GFP in a dilution of 1:1000 (α-GFP, ROCHE), followed by the incubation with an anti-mouse-HRP secondary antibody in a dilution of 1:4000 (GE Healthcare).

## Infection experiments

To assess virulence, we used the murine inhalation model of cryptococcosis [44]. For each strain, 10 female C57BL/6 mice were used. Mice were anesthetized by isoflurane inhalation and intranasally inoculated with 1 x 10$^5$ fungal cells of PBS-washed overnight cultures of indicated strains. Mice were monitored over the course of 40 days and sacrificed based on clinical endpoints that predict mortality. The statistical significance of difference between survival curves of mice infected with different strains was determined by log-rank test (GraphPad Prism). For CFU analysis, 5 mice per strain and time endpoint were intranasally inoculated as described above. Mice were sacrificed at indicated endpoints post infection and the mouse lung harvested, weighed, and homogenized in cold PBS. Colony forming units (CFU) were calculated by quantitative culture and are represented as CFU/gram wet weight. For immunohistochemistry at 3 days post infection, one side of infected mice lungs was fixed in buffered formalin and submitted to the Immunohistopathology Core at Duke University.

To assess virulence at lower temperature, we used *Galleria mellonella* larvae as an invertebrate model of cryptococcosis. Larvae were purchased at Vanderhorst in St. Marys, Ohio (waxworms.net). For each incubation temperature and cryptococcal strain, we infected 30 larvae by injecting 10$^7$ cells into the protopod of each larva. Larvae were monitored over the course of infection daily (melanization, movement). Larvae transitioning to the pupa state were excluded from the study. For analysis, survival data were plotted using a Kaplan Meier Plot and statistics were performed using the Log-rank test (GraphPad Prism).

## Arabidopsis thaliana infection protocol

**Fungal growth condition.** *C. neoformans* WT, *cel1Δ*, and *cel1Δ*$^C$ were sub-cultured twice in yeast peptone dextrose (YPD) broth at 30°C, shaking at 150 rpm. Cells were collected by

centrifugation, washed three times in ddH$_2$O, and re-suspended to OD$_{600}$ = 0.1 ($\approx 10^7$ cells/mL) in ddH$_2$O. *Arabidopsis thaliana* wild-type line Columbia Col-0 seeds was sterilized by washing first in 70% ethanol with 0.01% Triton X-100 for 5 min with regular mixing, and then in 95% ethanol for 5 min. The seeds were pipetted onto sterile Whatman paper to allow the ethanol to fully evaporate. The sterile seeds were sown on heat-treated soil (soil, sand, and vermiculite mixed in 3:1:1 ratio). To synchronize germination, the pots were covered with plastic film and stored for 3 days at 4°C in darkness. After synchronization, the plants were grown in a climate-controlled chamber (16 hr day, 200 µE × m$^{-2}$ × s$^{-1}$/8 hr night, 20°C, 70% relative humidity).

**Infection protocols.** Plant infection was carried out on 20–28 three week old plants using four different inoculation techniques: 1) Scarification (two leaves on each plant were scared with a sterile razor blade and 5 µL of a 10$^7$ cells/mL culture was laid on top), 2) Infiltration (the culture suspension was infiltrated on the backside of two leaves from each plant), 3) Spray (each plant was sprayed, using single use spray bottles to give a more severe overall infection of each plant), and 4) Drop inoculation (a 5 µL drop was inoculated onto two leaves from each plant). As controls, the same treatment of the plants was performed with sterile ddH$_2$O.

**Scanning electron microscopy.** Leaf pieces of approx. 5x5 mm were collected at 0 and 7 days after inoculation with *C. neoformans* and fixed in 2.5% glutaraldehyde and 4% formaldehyde in 0.1 M cacodylate buffer, pH 7.2 for 1 h. The samples were then serially dehydrated with increasing concentrations of ethanol, and critical point dried with liquid CO$_2$ in an Electron Microscopy Sciences EMS 850 CP drier. Pieces of upper and lower leaf sides were mounted onto metal stubs with double-sided sticky tape and coated with gold: palladium (1:1) in a Polaron SC 7640 (Quorum Technologies, Newhaven, UK) automated sputter coater. Observations of plant and fungal structures were made from 3 leaves of each experiment using a FEI Quanta 200 ESEM (FEI Company) operated at 10 kV.

## Scanning electron microscopy (SEM)

Prior to SEM analysis, cells were conditioned as indicated. Cells were harvested and fixed for 1h at RT with 2.5% glutaraldehyde in PBS, washed 3 times with PBS and checked for intact cells and, if possible, capsule by India ink staining. Then, cells were mounted onto poly-L-lysine-coated coverslips (Neuvitro, 12mm, #1 thickness coverslips) and incubated for 20 min at RT. After mounting, cells were sequentially dehydrated in several ethanol washes (1x 30%, 1X 50%, 1X 70%, each 5 min RT, followed by 1x 95% and 2x 100%, 10 min RT). After dehydration, mounted cells were stored in 100% ethanol until the critical point drying. Cell samples were critical point dried with a Tousimis 931 critical point dryer (Rockville, Maryland) and coated with gold-palladium using a Cressington 108 sputter-coater (Watford, United Kingdom). Samples were mounted and imaged on a Hitachi S-4700 scanning electron microscope (Tokyo, Japan).

## Capsule electromobility assay

Method was based on Yoneda *et al.*, 2008 [96]. Briefly, from a single colony on YPD plates, an overnight culture was started in YPD broth at 30°C, 150 rpm. The overnight culture was adjusted to OD$_{600}$ of 1.0 and 500 µL was used to inoculate 5 mL CO$_2$ independent medium (Gibco, Invitrogen, MA, USA) supplemented with 10% v/v heat inactivated fetal bovine serum (Thermo Fisher Scientific, IL, USA). The culture was incubated for three days at 37°C, 150 rpm. The culture was spun down at 3000 x g for 3 min. 1.5 mL of the supernatant was sterile filtered through a 0.22 µm PES filter (Sarstedt, Nümbrecht, Germany). 15 µL of sterile supernatant was mixed with 2.5 µL 6X DNA loading dye. Samples were loaded on a 0.6% w/v agarose

gel (9 cm long) and separated for 15 hours at 25 V. The samples were transferred to a positively charged membrane using southern blot protocol from the Whatman Nytran SuPerCharge TurboBlotter kit (Whatman GE Healthcare, Little Chalfont, UK). Air dry the membrane, block with 5% w/v skim milk in TBS-T for one hour, followed by immunoblotting with 1:1000 anti-GXM (mAB 18B7 (mouse), Merck, MA, USA) for one hour in 5% skim milk in TBS-T. Washed three times with TBS-t, and then treated with anti-mouse secondary antibody HRP-conjugated for one hour in 5% skim milk in TBS-T. The blot was washed three times 5 min with TBS-T, before it was developed with SuperSignal West Dura Extended Duration substrate (Thermo Fisher Scientific, IL, USA). The blot was imaged using a Bio-Rad Chemidoc XRS+ (Bio-Rad, CA, USA).

## Cell wall staining and microscopy

Prior to analysis cells were conditioned as indicated. Then, cells were harvested and stained with 100 μg/ml Alexa488-conjugated wheat germ agglutinin (WGA, Molecular Probes) for 35 minutes in the dark, RT, followed by the addition of 25 μg/ml calcofluor white (CFW, Fluka Analytical). Cells were incubated for 10 minutes, RT. Then, cells were washed 2x with PBS and were resuspended in 20–50 μL PBS for microscopic analysis. The Alexa488-WGA signal was imaged using a GFP filter and CFW signal was imaged using a DAPI filter. Images were further processed and analyzed using Fiji/ImageJ.

## Cell wall isolation and analysis

Prior to cell wall analysis, cells were conditioned as indicated. Chitin and chitosan levels were quantified from lyophilized yeast using a modified MBTH (3-methyl-benzothiazolinone hydrazine hydrochloride) method as previously described [4]. β-glucan was quantified using the megazyme yeast β-glucan kit as previously described [4].

## Microscopic quantification

Differential interference microscopy (DIC) and fluorescent images were visualized with a Zeiss Axio Imager fluorescence microscope (64X objectives). Images were taken with an Axio-Cam MRm digital camera with ZEN Pro software (Zeiss). The same exposure time was used to image for every strain analyzed in one data set. Images were analyzed using ImageJ/Fiji software. For indicated strains a DAPI stain (NucBlue, Invitrogen) was used to visualize the nucleus. DAPI stained cells were analyzed using the DAPI stain, Cel1-GFP were visualized using the GFP channel. Cells sizes were measured using the ImageJ measurement tool. Capsule thickness was calculated using the equation:

$$capsule\ thickness = \frac{(cell\ diameter\ including\ capsule - cell\ body\ diameter)}{2}$$

Histopathology images were taken with a color camera (AxioCam MRm) attached to the Zeiss Axio Imager and analyzed using ImageJ/Fiji.

## Determination of cell size and budding index

For each experiment, cells were grown overnight in liquid YDP medium at 30˚C. The cells were diluted in fresh YDP medium and incubated for and additional 2 h. The cells were harvested and resuspended in either YPD or CIM medium ($OD_{600}$ 0.2). The cultures were incubated for 18h at either 30˚C (YPD) or 37˚C (CIM), fixed, and briefly sonicated before microscopic analysis. Microscopic images were analyzed using ImageJ/Fiji. Cells were

counted, and the ratio between non-budded and budding cells was calculated. Cell size was measured using the Image J measurement tool.

## Propidium iodine (PI)-based Ploidy analysis and flow cytometry

Prior to analysis cells were treated as indicated. For propidium iodine staining cells were treated as described in Altamirano *et al.* [78]. In short, cells were harvested, washed and fixed with 70% ETOH at RT for 1h and subsequent incubation at 4˚C ON. Subsequently, fixed cells were washed and incubated with 1 μL RNAse A (in 500 μL of RNAse A Buffer: 0.2 M Tris pH 7.5, 20 mM EDTA) for 4h at 37˚C. Then cells were harvested and washed, and PI staining was performed using 0.05 mg/mL PI in 100 μL in PBS for 30 minutes in the dark at RT. Samples were washed twice with PBS and submitted for FACS analysis. The FACS analysis was performed at the Duke Cancer Institute Flow Cytometry Shared Resource using a BD FACSCanto II flow cytometer. Data was analyzed using FlowJo v10.1 software (FlowJo, LLC). For ploidy analysis histograms with mean fluorescence intensity (MFI) on the x-axis and cell counts on the y-axis were created. Unstained cells were used as negative controls.

## Minimal inhibitory concentration (MIC) analysis for cinnamon extract

MIC analysis was performed in either unbuffered YNB or YNB buffered with 150 mM HEPES to pH 7.4 (YNB-pH 7.4). The methanol based cinnamon extraction was performed as described in [40]. The extract was dissolved in 50% DMSO for further analysis (in text referred to as cinnamon extract). The MIC experiment was performed with a 2-fold serial dilution of cinnamon extract (concentration range 313–1.0 μg/mL). Initial cell concentrations for each condition was ~500 cells/mL. A growth control, with no extract treatment, was included for each strain and medium. All conditions and treatments were prepared with eight biological replicates. The experiment was performed in 96-well plates at 30˚C for 72 hrs. Growth graphs of the indicated strains, at the conditions analyzed, were generated by calculating the relative growth of the drug-treated condition in relation to the untreated condition (treated $OD_{530nm}$/ untreated $OD_{530nm}$). As a control, the same protocol was used to evaluate inhibition of growth from the highest DMSO concentration used.

## Statistical analysis

All data error bars represent statistical errors of the means (SEM) of results from a number of biological replicates (N), as indicated in figure legends. Before statistical analysis was conducted, data from all experiments was log transformed for comparison of proportions. Statistical analysis was performed with GraphPad Prism software v9. The statistical tests chosen for each experiment and their results (i.e., p values) are indicated in figure legends. Asterisks in figures correspond to statistical significance as follows: ****, $P < 0.0001$; ***, $P = 0.0001$ to $P < 0.001$; **, $P = 0.001$ to $P < 0.01$; *, $P = 0.01$ to $P < 0.05$; ns (not significant), $P > 0.05$.

## Supporting information

**S1 Fig. (A)** Schematic domain overview of *Cn*Cel1 protein variants. The N-terminal signal peptide is highlighted in red, putative copper binding amino acids are highlighted in blue and labelled, and the unstructured C-terminus is highlighted in dark green. **(B)** Structure based alignment of Cel1 and the AA9 sequences from the phylogenetic analysis. The alignment was created using EXPRESSO of the T-COFFEE alignment package [85]. The final alignment was prepared using the ESPript 3.0 web server [90]. The substrate binding regions, L2, L3, and LC, are indicated, as well as the two paired cysteine disulfide bridges and two unpaired cysteines

from the two protein fold models.
(TIF)

**S2 Fig. (A)** Alphafold2 model of *Cn*Cel1, turned 180 on the y-axis compared to Fig 1B. *Cn*Cel1 is shown in magenta. *Cn*Cel1 aromatic residues potentially involved in substrate binding are shown as stick representation in blue and *Cn*Cel1 cysteines in yellow. The two unpaired cysteines are indicated. **(B)** Representative coomasie stained polyacrylamide gel of the expression supernatant, purified, and deglycosylated $Cel1_{Cat}$ protein. **(C-D)** Quantitative amino acids acid analysis of deglycosylated $Cel1_{Cat}$ confirms expression of correct protein. **(E)** Progress curves for oxidation of reduced fluorescein by non-enzyme bound copper and purified $Cel1_{cat}$ without added copper. Activity measured with relative fluorescent units (RFU) at 528 nm for 30 min. Free $CuCl_2$ was tested as a negative control in stoichiometric amounts to a 5 μM copper loaded $Cel1_{cat}$ (3.75 μM). $Cel1_{cat}$ concentrations as indicated, without copper loading, was tested to assess initial copper loading from expression host. Reaction conditions are 75 mM phosphate citrate, pH 7.4, 25°C, $Cel1_{cat}$ blank (black line), 100 μM DHA (red line), 100 μM $H_2O_2$ (blue), or both (green line) was added to investigate LPMO copper reduction. Experiments done in triplicates, standard deviations shown, but not visible.
(TIF)

**S3 Fig. (A)** Phenotypic analysis of the Cel1-4xFLAG strain. The WT, *cel1Δ* mutant, and two Cel1-4xFLAG expressing strains were spotted in serial dilutions on YPD or YPD pH 8.15 medium and incubated for 3d at indicated temperatures. **(B)** Spotting based lung CFU analysis from lungs lysates 60 days post infections. Lungs were harvested from mice infected with the *cel1Δ* strain. Concentrated as well as a serial dilution of the harvested Lung lysate was spotted on YPD-Chloramphenicol and incubated for 3d at 30°C. **(C)** Plant infection analysis. 28 three week-old, soil-grown *Arabidopsis thaliana* col-0 plants were drop inoculated with either sterile water, WT, *cel1Δ*, or the *cel1Δ^C* strains. Representative images are shown of plants at the inoculation day (0 days post inoculation, dpi) and 7 dpi. Leaf samples infected with the WT strain were collected for scanning electron microscopy to examine fungal phenotype and behavior. White scale bar illustrates 50 μm for the right column and 10 μm for the left column, respectively. **(D)** Transcript analysis of *CEL1* abundance in *cel1Δ^{C-H17A}* and other indicated strains after a 3h induction in SC pH8.15 at 30°C. Quantitative RT-PCR was used to assess relative *CEL1* transcript levels for each strain compared to the WT strain. Presented is the mean +/- SEM of the relative transcript levels of 3 biological replicates. **(E)** Five-fold serial dilutions of cell suspensions for each strain were incubated on YPD or YPD pH 8.15 at 30°C or 37°C for 3d, in the presence or absence of lysing enzymes extracted from *Trichoderma harzianum*. **(F)** Inhibition growth control of the highest DMSO concentration used in the MIC analysis at both pH conditions. Growth inhibition reported as in the MIC graph (G). **(G)** Minimal inhibitory concentration (MIC) analysis of cinnamon extract, dissolved in 50% DMSO, in unbuffered YNB and host condition-buffered YNB (YNB-pH 7.4). *Cn* WT and *cel1Δ* strains were cultivated in 96-well liquid cultures and treated with 2-fold serially diluted cinnamon extract. $OD_{530}$ was measured after 72h of growth at 30°C and the $OD_{530}$ of the untreated condition was set to 1.0. Presented is the mean +/- SEM of the relative growth ($OD_{530}$) of 8 biological replicates. **(H)** Corresponding cell wall chitosan levels of the chitin quantification shown in Fig 5D. Indicated strains were incubated for 24h in YPD 30°C, YPD pH 8.15 30°C or YPD 37°C. Cell wall material was purified from lyophilized yeast and chitosan was quantified using the MBTH-based chitin/chitosan quantification method. Data represent the mean +/- SEM of 4 biological replicates. A 2-way ANOVA was performed from log transformed data using GraphPad Prism.
(TIF)

**S4 Fig. DIC and single channel images of the merged microscopy images shown in Fig 4C.** The WT and *cel1Δ* strains were incubated overnight in YPD medium and resuspended to OD$_{600}$ of 0.1 in indicated medium (either YPD or YPD pH 8.15). Cells were cultivated for 24h hours at indicated temperatures and stained with CFW (total cell wall chitin) and WGA-Alexa488 (exposed cell wall chitin). Shown are the single channel images of CFW (blue channel), WGA-Alexa-488 (green channel) used for creating the merged images with Image J/Fiji as well as the corresponding DIC image of this analysis.
(TIF)

**S1 Table. *Cn* strains used in this study.**
(TIF)

**S2 Table. Plasmids used in this study.**
(TIF)

**S3 Table. Oligonucleotides used in this study.**
(TIF)

## Acknowledgments

We thank the Duke Cancer Institute for the use of the Flow Cytometry Shared Resource. Scanning electron microscopy was performed at the Chapel Hill Analytical and Nanofabrication Laboratory, CHANL, a member of the North Carolina Research Triangle Nanotechnology Network, RTNN. We thank Dr. Dennis Thiele for the initiation of the collaborative work between laboratories at Duke University and Copenhagen University. We thank Dr. Bin Li for his help with conducting the FACS analysis. We thank former PhD student at University of Copenhagen Kasper Sjödin for his initial discovery of the *cel1Δ* rough colony phenotype, which propelled this study and Dr. Radina Tokin for providing cinnamon extract for our MIC analysis.

## Author Contributions

**Conceptualization:** Corinna Probst, J. Andrew Alspaugh.

**Data curation:** Corinna Probst, Magnus Hallas-Møller, Johan Ø. Ipsen, Jacob T. Brooks, Karsten Andersen, Helle J. Martens, Connie B. Nichols.

**Formal analysis:** Corinna Probst, Magnus Hallas-Møller.

**Funding acquisition:** Katja S. Johansen, J. Andrew Alspaugh.

**Investigation:** Corinna Probst, Magnus Hallas-Møller, Johan Ø. Ipsen, Jacob T. Brooks, Karsten Andersen, Mireille Haon, Helle J. Martens, Connie B. Nichols.

**Methodology:** Johan Ø. Ipsen, Jacob T. Brooks, Mireille Haon, Jean-Guy Berrin, Helle J. Martens.

**Resources:** Mireille Haon, Jean-Guy Berrin.

**Supervision:** Katja S. Johansen, J. Andrew Alspaugh.

**Writing – original draft:** Corinna Probst, Magnus Hallas-Møller, Jean-Guy Berrin, Katja S. Johansen, J. Andrew Alspaugh.

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
