## [Decision Letter · Decision Letter 0]

1 Dec 2022

Dear Dr. Alspaugh,

Thank you very much for submitting your manuscript "A fungal lytic polysaccharide monooxygenase is required for cell wall integrity, thermotolerance, and virulence of the fungal human pathogen Cryptococcus neoformans" for consideration at PLOS Pathogens. As with all papers reviewed by the journal, your manuscript was reviewed by members of the editorial board and by several independent reviewers. In light of the reviews (below this email), we would like to invite the resubmission of a significantly-revised version that takes into account the reviewers' comments. Specifically, conclusions related to the protein's sub-cellular localization and function are not currently rigorously supported by the presented data. A major question from all three reviewers related to the substrate involved in the protein's function, at least ruling out some potential candidate, and the localization studies. Some controls are missing to help strengthen conclusions, raised by reviewer 1, for example functionality of GFP tagged alleles. All reviewers also appreciated the very nice phenotypic characterization of the mutant strains and important virulence result. With some additional insights into the sub-cellular localization and function of the protein, the study is an impactful contribution to the field. Addressing both substrate specificity and sub-cellular localization is likely beyond the scope of a revision, but from my perspective definitely confirming the localization is likely to yield testable hypotheses about potential substrate(s) for follow up studies. Alternatively, the author's could take advantage of the recombinant protein and test well known substrates. 

We cannot make any decision about publication until we have seen the revised manuscript and your response to the reviewers' comments. Your revised manuscript is also likely to be sent to reviewers for further evaluation.

Sincerely,

Robert A. Cramer, PhD

Academic Editor

PLOS Pathogens

Alex Andrianopoulos

Section Editor

PLOS Pathogens

Kasturi Haldar

Editor-in-Chief

PLOS Pathogens

orcid.org/0000-0001-5065-158X

Michael Malim

Editor-in-Chief

PLOS Pathogens

orcid.org/0000-0002-7699-2064

Reviewer's Responses to Questions

**Part I - Summary**

Reviewer #1: Probst et al. report the characterization of a lytic polysaccharide monooxygenase Cel1 and the phenotypes of mutants lacking this enzyme for C. neoformans. Key findings indicate that the enzyme is expressed in a pH and temperature dependent manner, and that it is localized to the cell wall. Mutant analysis revealed contributions to temperature and cell wall stress responses. An impact on cell cycle progression was also noted. Importantly, Cel1 was also important for virulence in mouse and Galleria models of disease. Overall, the study is well presented, interesting and informative in the context of connections between cell wall integrity, virulence, and cell cycle progression. Another strength is the novelty in the focus on LPMOs in an animal pathogen. There were a number of weaknesses including the depth of the analysis (substrate identification) and the interpretation of some of the results. I have specific comments and concerns, as listed below.

Reviewer #2: In this study, the authors investigate a LPMO in C. neoformans and find the protein is essential for virulence and for cell wall integrity through an unknown mechanism. The study for the most part is rigorous and logical and the manuscript is written clearly for a diverse audience. Although these studies clearly demonstrate an importance for this protein in virulence, the underlying mechanism of LPMO action or possible substrates were not explored.

Reviewer #3: In this paper, Probst et al report on a presumed lytic polysaccharide monooxygenase (LPMO) of C. neoformans. They show that expression of the corresponding gene is increased under host physiological conditions and that it is required for stress tolerance of the fungi. They further suggest that the protein is localized to the cell wall and acts in intrinsic cell wall remodeling events required for survival in the host. Overall, the description of this protein is straightforward and the experiments showing characterization of protein expression and mutant phenotypes are particularly well performed and presented. The conclusions about function, however, while potentially an interesting contribution to the important and understudied area of cell wall synthesis, are somewhat less compelling (see below).

**Part II – Major Issues: Key Experiments Required for Acceptance**

Reviewer #1: 1.Line 97. The hypothesis to justify focusing on LPMOs (endogenous fungal substrates) is presented, but the substrates are not identified in the initial section of the Results (e.g., lines 146-155; Discussion line 420-421). The manuscript would be stronger if substrates were identified and linked to the subsequent focused analysis of the cell wall.

2. Line 162, Fig. 2A. The expression of the flag-tagged protein is remarkably strong and specific for the pH 8 condition and no signal (even a weak one) is seen for the other conditions/lanes. Can the authors comment on why the expression is so specific?

3. Line 169. No expression is seen in the rim101 mutant. However, in Fig. 5A there is a difference in the phenotypes for the rim101 mutant versus the cel1 mutant at 37C – e.g. on SDS, CFW, caffeine. What is the explanation for this difference – the media conditions for the experiments?

4. Line 193. Is the cel1ΔMT1-Cel1-GFP construct functional?

5. Line 196. The evidence is not clear for the conclusion in title for Fig. 3 that trafficking occurs by the ER/Golgi route – a more detailed analysis would be needed to support this conclusion - eg. BFA treatment. The images in Fig. 3C are a bit murky (especially the DAPI) and it would help to have DIC images of the cells.

6. Lines 246-261. This section reports negative results and seems strangely uninformative about Cel1 given that the wt strain did not cause symptoms. Why would one expect yeast cells to penetrate the plant surface? In general, I think this section detracts from the paper and could just be mentioned as a brief statement.

7. Line 295. Is it known that the mutated Cel1-H17A protein was actually expressed?

8. Lines 310-311, Figure 5C. It was difficult to appreciate that the evidence from the figure justified the conclusion. It would help to have DIC images for clearly identifying the multiply budded cell sites, and perhaps a more detailed analysis of staining would demonstrate the difference. Could this be quantified by flow cytometry?

9. Lines 338-350. In general, my examination of Fig. 6B,C,D and F suggests not obvious difference for the capsule. The enlarged cell body is the only convincing result.

10. Lines 392-396. This first paragraph contains a jumble of ideas (especially the first sentence) and could be deleted.

11. Line 537. Did the authors consider testing cinnamtannin B1 (if available) with C. neoformans?

Reviewer #2: 1) Although the authors have the enzymatically active recombinant protein, they did not test this against any candidate substrates of LMPO, e.g., chitin or other polysaccharides in the cell wall or capsule. Even a negative result would help to pinpoint the mechanism by ruling out possibilities.

2) The localization studies were conducted with a protein greatly over-expressed under Cu toxicity stress conditions. They find much of the protein in the Golgi and ER. Is it being trapped there because of the over-expression and/or Cu stress conditions? The authors claim there is some cell surface localization, but Cel1 localization looks indistinguishable from the control Sec63 that is ER/Golgi. If Cel1 is expressed at physiological levels, would it largely show localization at the cell surface where it can modify the capsule or cell wall material? The accurate localization of this protein would certainly shed important light into the mechanism of action.

Reviewer #3: Major Issues. *Note that only III would constitute key data required to validate conclusions.

I. The authors show that Cn CEL1 is an outlier in the AA9 group and that the patterns of oxidation Cel1 catalyzes are quite different from those of a control AA9 protein (Fig. 1D). Are they certain it has the attributed activity? This figure is also central to the conclusions of the paper, so enough information must be provided for the reader to understand and evaluate these results. For example, the text for Fig. 1D should include the rationale for adding ascorbate, hydrogen peroxide, or both. It would also be helpful to know why fluorescein was used instead of the substrate characterized in the cited paper (22).

II. Despite having achieved expression and purification of an active protein, the authors delve no further into the biochemical function of Cel1. Given the major questions about wall synthesis that (as they point out) remain to be answered, this seems like a missed opportunity. Further analysis might also address the previous point.

III. Central to the authors’ conclusion is that Cel1 remains at the cell surface. The very nice images in Fig. 3C clearly show that Cel1-GFP occurs in the secretory pathway, but this could also occur if it was en route to release from the cell. The key information, therefore, is the difference between the cellular distribution of Cel1-GFP and the Sec63-GFP control. While the first image presented for Cel1-GFP appears to show puncta on the cell surface that are not present with Sec63-GFP, the second Cel1-GFP image looks much like the control. The authors should quantitate surface versus internal puncta or otherwise further support this key critical conclusion. The Discussion should also include how the authors think Cel1 remains associated with the cell surface.

Other issues that would potentially require experimentation (with line number):

311 The statement about WGA staining intensity is not supported by the images shown and should be modified or supported by additional data.

348 Fig. 6F is hard to interpret because of apparent variability in this method. For example, the first and third lanes look more similar than the first and second. Although the mutant material does run slightly higher on the gel, its most intense region, at least in lane three, is lower than WT. Overall, the conclusions about this experiment do not seem well supported and should be modified or supported by additional data.

**Part III – Minor Issues: Editorial and Data Presentation Modifications**

Reviewer #1: I focused on key issues, as presented in Part II.

Reviewer #2: 1) Previous studies by Thiele et al described a LMPO like protein Bim1 that is important for Cu homeostasis but there is no mention of this in the introduction.

2) In Fig. 2A the authors only tested one dose of BCS to conclude Cel1 is not regulated by Cu starvation. Is there a control to show that this dose creates a Cu starvation stress state in the yeast?

3) Is the control localization of Sec63 in Fig. 3 also done under Cu toxicity stress conditions?

4) The studies attempting to show fungal virulence in a plant model (without success) seem beyond the scope of this paper.

Reviewer #3: 81 How carbohydrates might be ‘recalcitrant’ should be explained.

116 The authors should mention why they narrowed their focus to 00601.

130 What constitutes ‘acceptable’ should be explained.

145 What stain is used in Supp Fig 2B?

161 ‘Phenotypically identical’ is an overstatement given that growth in only two conditions is tested.

180 The differences in protein expression at various temperatures are modest, and the amount at 37° seems slightly less than at either 30° or 39°.

210 Although it does not change conclusions about Cel1, it is surprising that all mice did not succumb to WT/complement infection with the given inoculum.

272 Fig. 5A suggests the occurrence of suppressors. Have the authors looked into this?

294 The legend to Supp. Fig. 3 has errors including panel lettering. Also, there is no reason to show phenotypes of transformants that were incorrect and not used in the study.

PLOS authors have the option to publish the peer review history of their article (what does this mean?). If published, this will include your full peer review and any attached files.

Reviewer #1: No

Reviewer #2: No

Reviewer #3: No
---

## [Decision Letter · Decision Letter 1]

10 Apr 2023

Dear Dr. Alspaugh,

We are pleased to inform you that your manuscript 'A fungal lytic polysaccharide monooxygenase is required for cell wall integrity, thermotolerance, and virulence of the fungal human pathogen Cryptococcus neoformans' has been provisionally accepted for publication in PLOS Pathogens.

Best regards,

Robert A. Cramer, PhD

Academic Editor

PLOS Pathogens

Alex Andrianopoulos

Section Editor

PLOS Pathogens

Kasturi Haldar

Editor-in-Chief

PLOS Pathogens

orcid.org/0000-0001-5065-158X

Michael Malim

Editor-in-Chief

PLOS Pathogens

orcid.org/0000-0002-7699-2064

Reviewer Comments (if any, and for reference):

Reviewer's Responses to Questions

**Part I - Summary**

Reviewer #1: The authors have adequately addressed my concerns about the manuscript. In particular, I appreciated the addition of the MIC tests with cinnamtannin B. I have no additional concerns.

Reviewer #2: The authors have adequately addressed all concerns of the previous review and no additional revisions are required.

Reviewer #3: The authors have thoughtfully responded to reviewer comments, including revising their approach to establishing whether Cel1 is associated with the cell wall and addressing other matters.

**Part II – Major Issues: Key Experiments Required for Acceptance**

Reviewer #1: None

Reviewer #2: no issues

Reviewer #3: (No Response)

**Part III – Minor Issues: Editorial and Data Presentation Modifications**

Reviewer #1: None

Reviewer #2: no issues

Reviewer #3: (No Response)

PLOS authors have the option to publish the peer review history of their article (what does this mean?). If published, this will include your full peer review and any attached files.

Reviewer #1: No

Reviewer #2: No

Reviewer #3: No

---

## [Editor Report · Acceptance letter]

20 Apr 2023

Dear Dr. Alspaugh,

We are delighted to inform you that your manuscript, "A fungal lytic polysaccharide monooxygenase is required for cell wall integrity, thermotolerance, and virulence of the fungal human pathogen Cryptococcus neoformans," has been formally accepted for publication in PLOS Pathogens.

Best regards,

Kasturi Haldar

Editor-in-Chief

PLOS Pathogens

orcid.org/0000-0001-5065-158X

Michael Malim

Editor-in-Chief

PLOS Pathogens

orcid.org/0000-0002-7699-2064